# Relative timescale of channel voltage dependence and channel density regulation impacts assembly and recovery of activity

**Yugarshi Mondal[1]\*, Ronald L Calabrese[2], Eve Marder[1]**

[1]Volen Center and Biology Department, Brandeis University, Waltham, United States; [2]Biology Department, Emory University, Atlanta, United States

## eLife Assessment

This **important** computational study investigates homeostatic plasticity mechanisms that neurons may employ to achieve and maintain stable target activity patterns. The work extends previous analyses of calcium-dependent homeostatic mechanisms based on ion channel density by considering activity-dependent shifts in channel activation and inactivation properties that operate on faster and potentially variable timescales. The model simulations **convincingly** demonstrate the potential functional importance of these mechanisms.

**\*For correspondence:**
ymondal@brandeis.edu

**Competing interest:** The authors declare that no competing interests exist.

**Abstract** Neurons can maintain stable activity profiles over their lifetimes despite ion channel turnover over minutes to weeks. Neuronal activity is also influenced by regulating the voltage dependence of ion channels. How do these two forms of plasticity work together to maintain a stable activity profile? We augment a classical model of activity-dependent ion channel density regulation with a mechanism that adjusts channel voltage dependence based on activity. These findings reveal that the timescale of these mechanisms shapes the specific electrical activity patterns that achieve a target activity profile. Moreover, alterations in channel voltage dependence can impact a neuron's ability to recover from perturbations. These results highlight a potentially distinct role for activity-dependent regulation of channel voltage dependence in maintaining stable neuronal activity profiles.

## Introduction

Neurons maintain stable activity patterns over years despite continual turnover in their ion channel composition, which occurs on timescales from days to weeks. Experimental studies have shown that neurons homeostatically regulate both channel number (*Desai et al., 1999*; *Haedo and Golowasch, 2006*; *Mizrahi et al., 2001*; *Santin and Schulz, 2019*; *Thoby-Brisson and Simmers, 2002*; *Turrigiano et al., 1994*; *Turrigiano et al., 1995*; *Viteri and Schulz, 2023*) and channel voltage dependence (*Gasselin et al., 2015*; *Haedo and Golowasch, 2006*; *Thoby-Brisson and Simmers, 2002*) to stabilize their activity.

These mechanisms have been widely studied through computational modeling, but most of this work has focused on changes in channel number (*Abbott and LeMasson, 1993*; *Alonso et al., 2023*; *Golowasch et al., 1999*; *Gorur-Shandilya et al., 2020*; *LeMasson et al., 1993*; *O'Leary et al., 2014*; *Siegel et al., 1994*; *Srikanth and Narayanan, 2015*). This is largely due to the plethora of work indicating such changes occur (*Desai et al., 1999*; *Haedo and Golowasch, 2006*; *Mizrahi et al., 2001*;

*Santin and Schulz, 2019*; *Thoby-Brisson and Simmers, 2002*; *Turrigiano et al., 1994*; *Turrigiano et al., 1995*; *Viteri and Schulz, 2023*). However, this emphasis also reflects the challenges associated with precisely measuring changes in channel voltage dependence. Inter-neuronal variability in activation and inactivation curves can obscure clear patterns of activity-mediated modifications, especially when analyzed at the population level (*Turrigiano et al., 1995*). As a result, our understanding of how modifications to channel voltage dependence contribute to activity stability remains limited.

This gap is important to address given the ubiquity of channel voltage-dependence regulation. Neurons modulate voltage dependence through mechanisms ranging from rapid post-translational modifications (*Fraser and Scott, 1999*; *Johnson et al., 1994*) or slower processes like subunit synthesis and degradation (*Wu et al., 2023*). These processes are known to be associated with calcium-mediated second messengers. Intracellular calcium is also thought to play an important role in maintaining neuronal activity set points (*LeMasson et al., 1993*; *Turrigiano et al., 1994*). As such, calcium-mediated homeostasis may operate, at least in part, by tuning channel voltage dependence—yet the consequences of this form of regulation are largely uncharacterized.

To address this, we extended a state-of-the-art computational homeostatic model. This model stabilizes neuronal activity by monitoring intracellular $Ca^{2+}$ concentrations and inserting and deleting ion channels (*Alonso et al., 2023*). To this model, we added a mechanism that alters channel voltage dependence. Importantly, this model enabled independent control over the timescales of channel number and channel voltage-dependence alterations. This organization of the model let us simulate fast changes in voltage dependence, such as those driven by phosphorylation, or slower changes, like those resulting from synthesis or degradation of channel subunits. Our central finding is that the timescale of voltage-dependence regulation influences the specific channel configurations a neuron adopts—both while attaining a prespecified activity target and while maintaining it during perturbation.

## Results
### The model
To explore how modifications in channel voltage-dependence influence a neuron's capacity to achieve an activity profile as changes in ion channel density occur, we created a computational model. This model was built around a single compartment, Hodgkin–Huxley type conductance-based neuron with seven intrinsic currents (see *Turrigiano et al., 1995* or Appendix 1, Neuron model). The neuron model consisted of a fast $Na^+$ current, two $Ca^{2+}$ currents, three $K^+$ currents, a hyperpolarization-activated inward current ($I_H$), and a leak current. Each current was described by its maximal conductance (number of ion channels) and activation curves. Some currents also possessed inactivation curves as well. The (in)activation curves described the effects of voltage on the opening and closing of the membrane channels.

The activity-dependent regulation of channel number and (in)activation curves employed a strategy that uses intracellular $Ca^{2+}$ concentrations as a proxy for the neuron's activity (*Abbott and LeMasson, 1993*; *Alonso et al., 2023*; *Golowasch et al., 1999*; *Gorur-Shandilya et al., 2020*; *LeMasson et al., 1993*; *O'Leary et al., 2014*; *Siegel et al., 1994*; *Srikanth and Narayanan, 2015*). Specifically, we followed the approach used in *Liu et al., 1998* and *Alonso et al., 2023*, for altering channel density to guide our approach to altering (in)activation curves. In those models, the intracellular $Ca^{2+}$ concentration and three filters (also called sensors) of the $Ca^{2+}$ concentration were computed. The three sensors were employed to better differentiate between the activity of tonic spiking and bursting neurons (*Alonso et al., 2023*; *Liu et al., 1998*). A target activity profile was chosen by prespecifying the target values of these sensors. These values were selected through trial and error to consistently produce regular bursting across a specific range of initial conditions (see Methods). When deviations from these $Ca^{2+}$ sensor targets were detected, channel densities were altered to up- or down-regulate the number of channels in the membrane (*Alonso et al., 2023*; *Liu et al., 1998*). Details on how the sensors were computed and how the target activity—defined by the target sensor values—was specified are provided in the Methods.

We expanded on these models by incorporating a mechanism that shifts the (in)activation curves in response to deviations from the $Ca^{2+}$ sensor targets. This was accomplished by changing the half-(in)activation voltages of the (in)activation curves, mirroring strategies employed in *Liu et al.,*

*1998* and *Alonso et al., 2023*. The details of the Model, with their equations and assumptions, are described in the Methods section. A brief overview is provided in Appendix 1.

Importantly, the response times of the mechanisms responsible for ion channel insertion and deletion, as well as those for shifts in (in)activation curves, were independently controlled. In the model, these response times were initially set two orders of magnitude apart. This reflected the slower processes of protein synthesis and degradation associated with channel insertion and deletion. This contrasted with faster timescales of post-translational modifications such as channel phosphorylation. Specifically, the timescale for changes in maximal conductance was set at $\tau_g = 600$ s, and the timescale for half-(in)activation shifts at $\tau_{half} = 6$ s. Later, we adjusted the response times of the (in)activation curve shifts. In prior computational studies of activity-dependent regulation, $\tau_g$ was set to be less than 10 s, substantially faster than the timescales used in our model (*Alonso et al., 2023*; *Liu et al., 1998*). This study lengthens this timescale and explores the impact of varying the relative timescales of maximal conductance changes $(\tau_g)$ and half-(in)activation shifts $(\tau_{half})$.

## Bursters assembled by activity-dependent alterations in ion channel density and channel voltage dependence

*Figure 1* shows the full model, with the two regulation mechanisms in operation. From a set of starting randomly chosen parameters (maximal conductances and half-(in)activations), referred to as Starting Parameters 1 (SP1), the activity-dependent regulation mechanism altered the ion channel density and voltage dependences to assemble an electrical activity pattern that satisfied the Ca²⁺ targets. The targets were chosen so that a burster self-assembled (see Methods). The final ion channel density and voltage dependences are shown in the bar graphs (*Figure 1A*). The first two rows in *Figure 1A* show the electrical activity of the neuron as the model seeks to satisfy the Ca²⁺ targets. As it does, the model assembled a burster. The second row shows the activity over the entire trace, and the first row shows blows up at: the start (a), about 100 min (b), and at the end of the simulation (c). At the time shown in (b), the neuron was bursting, but the Ca²⁺ targets had not been achieved. There were two indications that this burster did not satisfy the specified Ca²⁺ targets: (1) the model still continued to adjust the properties of the ion channel repertoire here and (2) $\alpha$ was not yet zero.

Briefly, $\alpha$ measures whether the model has successfully tuned the neuron's ion channel densities and half-(in)activations to create an activity pattern that matches the preset sensor targets; values near zero indicate successful tuning (more in Methods). Even after the model assembles a channel repertoire that satisfies the targets, brief excursions of $\alpha$ away from zero can occur. These excursions exist because the neuron's electrical activity is quasistable—meaning it may lose/add a spike or experience a slight drop in slow wave amplitude, temporarily causing the sensors to be unsatisfied. When this happens, the model quickly makes minor adjustments to the channel densities and half-(in)activations to restore alignment with the targets (*Figure 1—figure supplement 1*, Row 2, between minutes 300 and 400). These adjustments are minimal (*Figure 1*, Rows 3–5).

The third and fourth traces in *Figure 1A* show how the model altered half-(in)activation and half-(in)activation values to meet the Ca²⁺ target. Very early on, there were large changes in the activation curves, which then slowly moved to a steady-state value. The fifth trace shows the evolution of maximal conductances. These started to change early on but did so slowly. As the maximal conductances slowly changed, the half-(in)activation curves followed suit, adapting to the new repertoire of ion channels to close the distance to the Ca²⁺ target. *Figure 1—figure supplement 1* shows the assembly of a burster from a different set of starting parameters (maximal conductances and half-(in)activations (SP4)), with qualitatively similar results to those obtained with SP1 but quantitatively different final parameters.

The results shown in *Figure 1A* require activity-dependent regulation of the maximal conductances. When activity-dependent regulation of the maximal conductances is turned off, the model failed to assemble SP1 into a burster (*Figure 1B*). This was seen in the other 19 starting parameters (SP2–SP20), as well.

In general, when the model was initialized with different starting parameters, it produced bursters with different final intrinsic properties. Twenty degenerate bursters (SP1–SP20), conforming to standards specified in Methods, were selected from over 100 bursters produced with different starting parameters. *Figure 2A* shows the activity characteristics of these 20 bursters (period, burst duration, spike height, maximum hyperpolarization, and slow wave amplitude). The left panel of

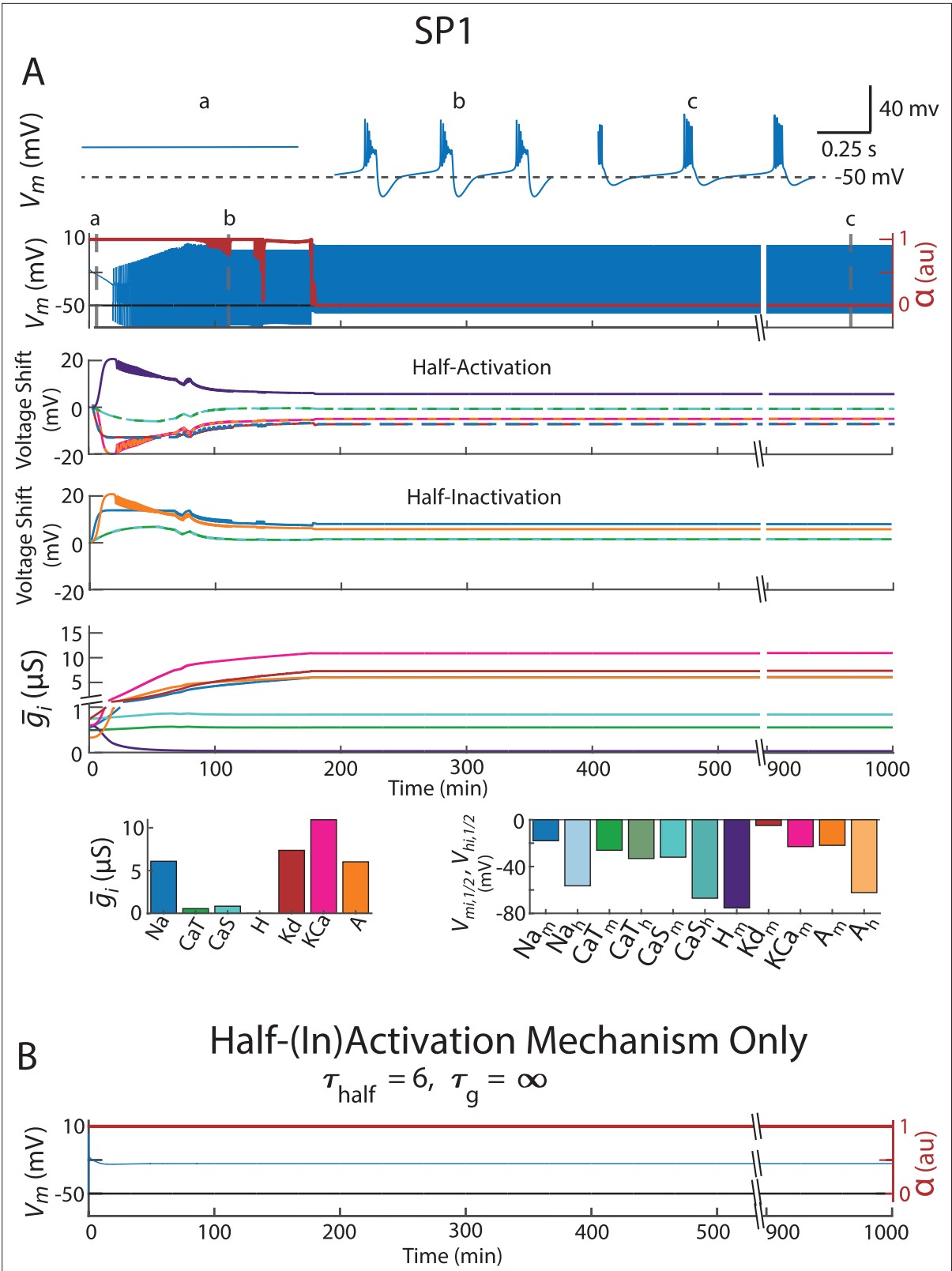

**Figure 1.** Activity-dependent regulation mechanism ($\tau_{half}$ = 6 s and $\tau_g$ = 600 s) was used to assemble a burster that meets the Ca²⁺ targets from 'random' starting parameters (i.e. SP). Both this set of starting parameters and its resultant model are called SP1. (**A**) Row 1: Blow-ups of specific points a, b, and c in the voltage trace in Row 2. Row 2: The neuron's electrical activity as its channel density and (in)activations curves were being adjusted. $\alpha$ is a measure of how close the model is to satisfying Ca²⁺ targets (see Methods). It took values between 0 and 1, with 0 meaning the targets were satisfied.

*Figure 1 continued on next page*

*Figure 1 continued*

Row 3: Plots the adjustments made to the half-activations as the model attempted to satisfy the calcium targets. They are color-coded to match the intrinsic currents shown in the left plot of Row 5. Row 4: Same as Row 3, except for (in)activation curves. Note that not all currents in this model have inactivation curves. Row 5: Same as Row 3, except for maximal conductances (i.e. channel density). Row 6: The final levels of maximal conductances are shown in the left plot. The final half-(in)activations are plotted on the right. The final half-activation of an intrinsic current, $V_{m_{i,1/2}}$, was the sum of the initially posited level of the that current's half-activation $V_{\hat{m}^0_{6,1/2}}$ and the shifts the model made to the activation curve, $V_{s^m_i}$ (shown in Row 3). The final half-inactivation of an intrinsic current, $V_{h_{i,1/2}}$, was the sum of the initially posited level of the that current's half-(in)activation $V_{\hat{h}^0_{i,1/2}}$ and the shifts the model made to the inactivation curve, $V_{s^h_i}$ (shown in Row 4). [The process is illustrated in *Figure 1—figure supplement 1* for a different set of starting parameters (i.e. initial half-(in)activation shifts and maximal conductance levels).] (**B**) This displays the neuron's electrical activity if the model was restricted to varying only half-(in)activations ($\tau_{half}$ = 6 s), with maximal conductances fixed ($\tau_g = \infty$ s). The same starting parameters as in Panel A were used for this simulation.

The online version of this article includes the following figure supplement(s) for figure 1:

**Figure supplement 1.** This model attempts to satisfy the same Ca²⁺ targets as in *Figure 1A*, but was started from a different set of 'random' initial parameters.

*Figure 2B* shows the range of conductances in this group. The right panel illustrates the variation in half-(in)activation curves. The asterisks indicate the baseline half-(in)activation values; all neuron models were assembled from starting parameters that lay within ±0.5 mV of these baseline values (see Methods for details).

## Half-(in)activation mechanism timescale impacts activity pattern

In the previous section, we studied the types of bursters that arose from slowly changing the density of the channel repertoire and quickly altering the half-(in)activations of the resulting repertoire. We next changed the timescale over which we altered half-(in)activations and assessed how the burst characteristics of the population changed (the timescale of channel density alterations remains the same). *Figure 3A* shows that alterations in the timescale of half-(in)activation alterations significantly impacted the period, interburst interval, spike height, and maximum hyperpolarization of the entire bursting population.

The timescale of half-(in)activation alterations also impacted some intrinsic properties of the population. *Figure 3B* plots the burster assembled with the timescale of half-(in)activation alterations at two extremes: fast and infinitely slow (i.e. off). This visualization is known as a currentscape (*Alonso and Marder, 2019*). In both cases, the resulting burster appeared to terminate using the same mechanism: hyperpolarization via the calcium-activated potassium current. This is seen by the relative contribution the calcium-activated potassium current makes at the point of maximum hyperpolarization (see at the dotted line in *Figure 3B*). However, the manner in which this current came to dominate the burster termination differs. Note that the burster constructed without half-(in)activation alterations possessed a higher maximal conductance for $I_{KCa}$ than the burster constructed with half-(in)activation alterations (*Figure 3C*, mauve dot). Moreover, the latter burster had more depolarized half-(in)activations (*Figure 3C*, mauve dot). In fact, this observation held true for SP2–SP20 as well (*Figure 3C*).

## Potential mechanism

To explore why the properties of the resulting bursters depend on the timescale of half-(in)activation adjustments, we examined what happens when SP1 is assembled under different half-(in)activation timescales: (1) fast, (2) intermediate (matching the timescale of ion channel density changes), and (3) infinitely slow (i.e. effectively turned off). The effects of these timescales can be seen by comparing the zoomed-in views of the SP1 activity profiles under each condition (*Figure 4*).

When half-(in)activations are fast, the time evolution of $\alpha$—which tracks how far the activity pattern is from its targets (see Methods)—shows an abrupt jump as it searches for a voltage-dependence configuration that meets calcium targets (*Figure 4A*). As this happens, the channel densities are slightly altered, and this process continues again. Slowing the half-(in)activations alterations reduces these abrupt fluctuations (*Figure 4B*). Making the alterations infinitely slow effectively removes half-(in)activation changes altogether, leaving the system reliant solely on slower alterations in maximal conductances (*Figure 4C*). Because each timescale of half-(in)activation produces a different channel repertoire at each time step, different timescales of half-(in)activation alteration led the model through

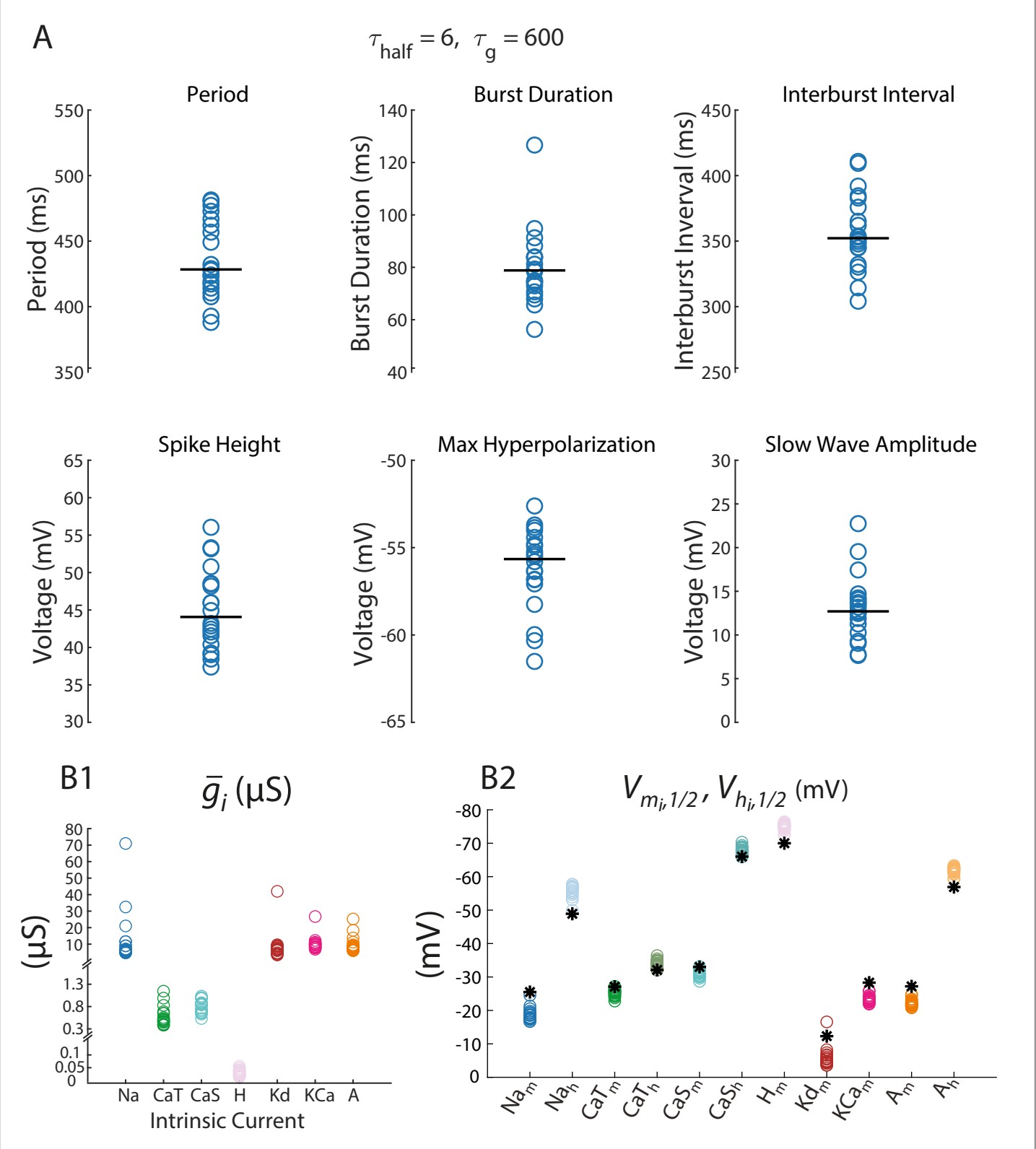

**Figure 2.** All 20 SPs were analyzed, including those illustrated in *Figure 1*. (**A**) The top left panel displays the periods for all SPs. Burst duration, interburst interval, spike height, maximum hyperpolarization, and slow wave amplitude measurements for these SPs are provided in subsequent plots. (**B**) B1 shows maximal conductances, and B2 displays half-(in)activation voltages for each of the 20 SPs. An asterisk on the right indicates the initially specified level of half-(in)activation from which random deviations were made.

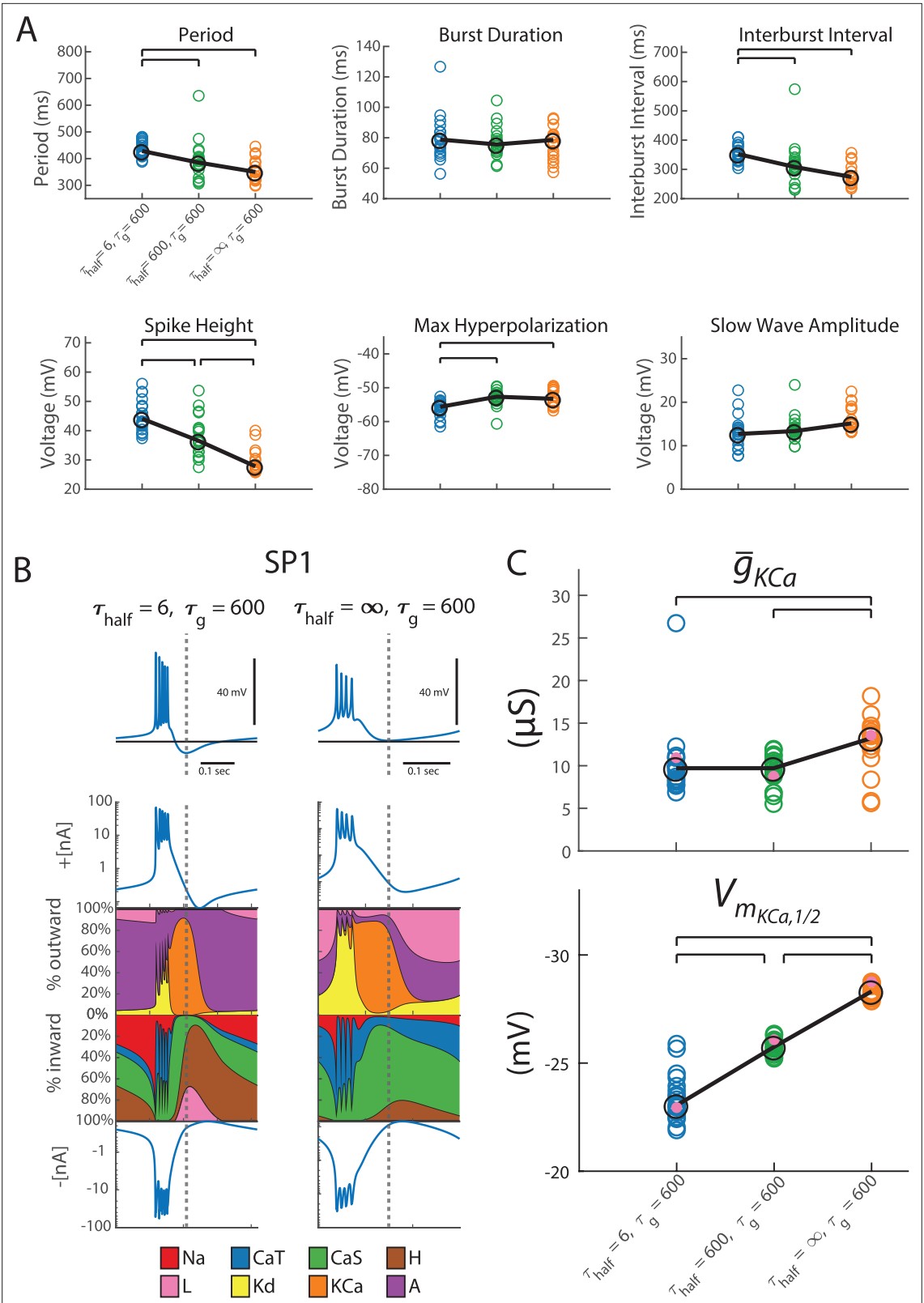

**Figure 3.** The speed at which the model changed half-(in)activations in response to deviations from Ca²⁺ target impacted the type of bursters the model assembled. In all simulations for this figure, $\tau_g$ = 600. (**A**) These plots extend the measurements from *Figure 2* to the same 20 SPs, now with half-(in)activations regulated at a different speed. In blue, $\tau_{half}$ = 6 s; recapitulating results in *Figure 2*. In green and orange, the SPs were reassessed with slower half-(in)activation adjustments, $\tau_{half}$ = 600 s, and $\tau_{half}$ = ∞ s (i.e. halted), respectively. Brackets indicate significant differences in medians (p

*Figure 3 continued on next page*

*Figure 3 continued*

< 0.05). These pairwise differences were assessed using a Kruskal–Wallis test (to assess whether any differences in medians existed between all groups) and post hoc Dunn tests with Bonferroni corrections to assess which pairs of groups differed significantly in their medians. (**B**) The model was used to stabilize SP1 to the same Ca$^{2+}$ target but using different timescales of half-(in)activation alterations. $\tau_{half}$ = 6 s is shown on the left and $\tau_{half}$ = ∞ s (representing the slowest possible response) on the right. Rows 1–4 below each voltage trace show: the total outward current, percentage contribution of each outward intrinsic current to the total outward current, percentage contribution of each inward intrinsic current to the total inward current, and the total inward current. A dashed vertical line across all rows marks the point of maximum hyperpolarization in each activity pattern to guide focused comparison. (**C**) When the model stabilized around a burster, the speed at which the model changed half-(in)activations impacted maximal conductance and half-activation location of the calcium-activated potassium current. Shown here in blue, green, and orange are $\tau_{half}$ = 6 s, $\tau_{half}$ = 600 s, and $\tau_{half}$ = ∞ s (i.e. halted), respectively. SP1–SP20 are in the colored circles. Trends are illustrated with a black line, with large circles marking median positions. The mauve dot highlights the levels to which maximal conductance and half-(in)activation of SP1 evolved. A Kruskal–Wallis test followed by post hoc Dunn tests (with Bonferroni corrections) was conducted to assess significant ($p < 0.05$) pairwise differences in medians (brackets).

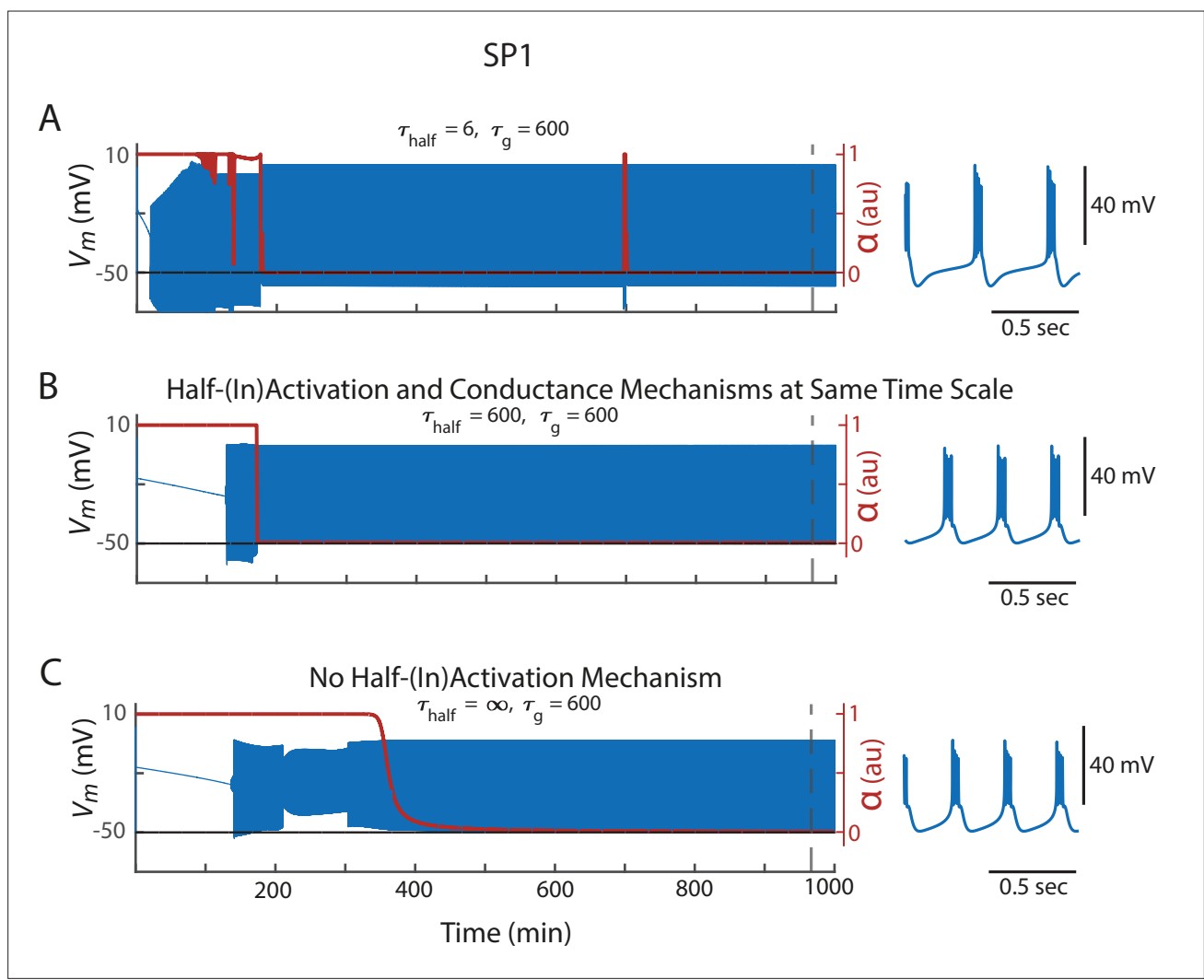

**Figure 4.** Changes to the timescale of half-(in)activation alterations steer the model through different trajectories in intrinsic parameter space. Each row shows the behavior of Starting Parameter 1 (SP1) as the model adjusts toward a configuration that satisfies the calcium activity target, using different timescales for half-(in)activation regulation. (**A**) $\tau_{half}$ = 6 s, (**B**) $\tau_{half}$ = 600 s, and (**C**) $\tau_{half}$ = ∞ s (i.e. half-(in)activations do not change). In all panels, the speed at which the model changed channel density is fixed in all rows: $\tau_g$ = 600 s. Superimposed on these traces is the $\alpha$ parameter, which indicates whether the activity patterns were meeting the Ca$^{2+}$ targets. Blow-ups of the activity pattern when the calcium sensors were satisfied are displayed to the right.

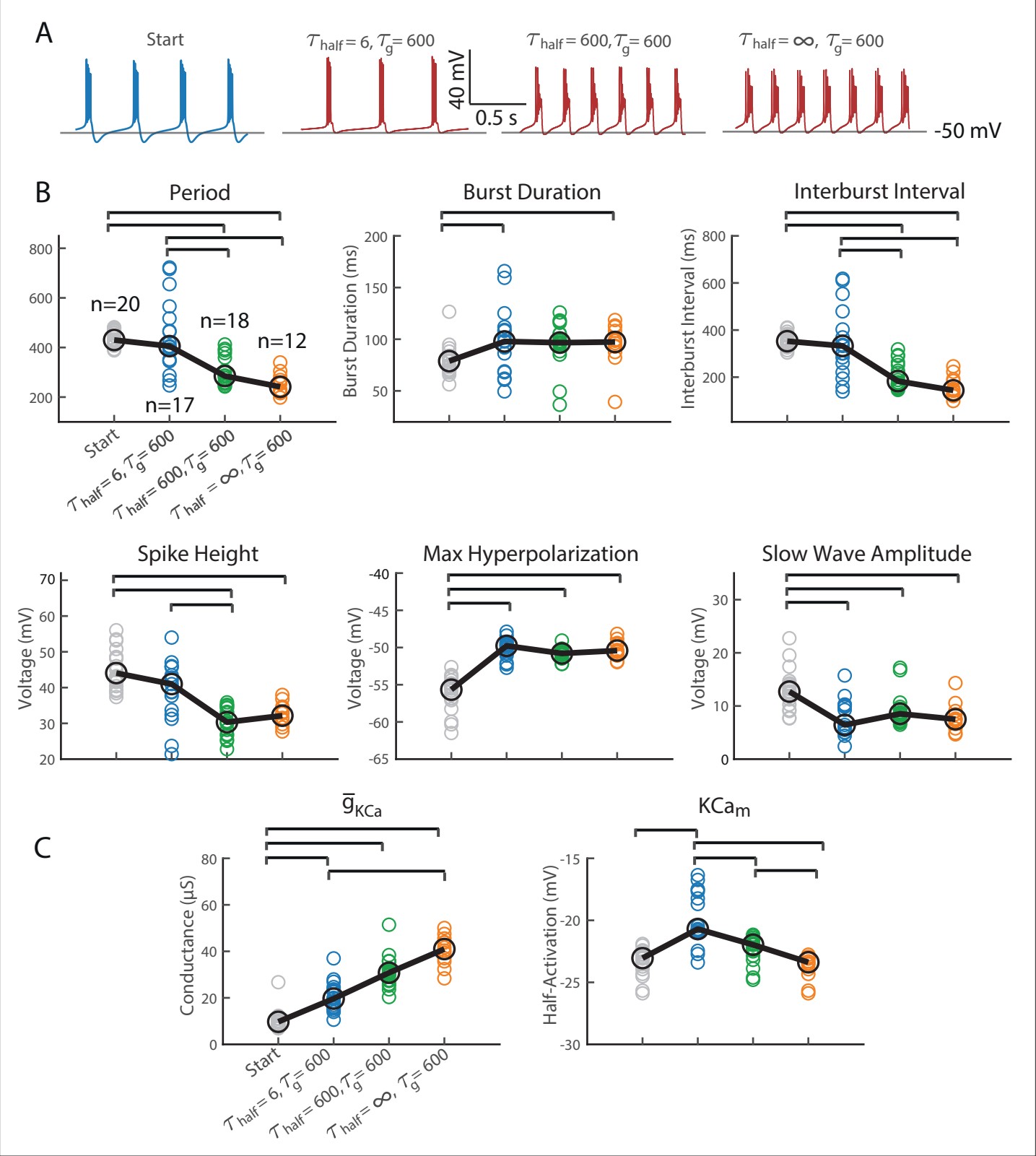

**Figure 5.** The speed at which the model changed half-(in)activations in response to deviations from Ca²⁺ targets impacted the type of bursters the model assembled following perturbation. (**A**) Voltage traces from SP1 before (blue) and 265 min after (red) a simulated increase in extracellular potassium concentration. Each panel shows recovery under a different timescale for half-(in)activation regulation: fast ($\tau_{half}$ = 6 s), intermediate ($\tau_{half}$ = 600 s), and infinitely slow (i.e. off, $\tau_{half}$ = ∞ s). In all conditions, alteration of maximal conductances was held fixed ($\tau_g$ = 600 s). (**B**) Summary of activity

*Figure 5 continued on next page*

*Figure 5 continued*

features for all SP1–SP20 models that recovered to a stable burst pattern. The number of models that successfully recovered varied with the homeostatic mechanism: fast ($n = 17$), intermediate ($n = 18$), and no half-(in)activation modulation ($n = 20$). Gray points indicate pre-perturbation measurements of SP1–SP20 (from *Figure 3A*). Significant pairwise differences in median values were assessed using a Kruskal–Wallis test followed by post hoc Dunn tests with Bonferroni correction. Significant comparisons ($p < 0.05$) are marked with horizontal brackets. (**C**) The speed of half-(in)activation regulation influenced both the maximal conductance and half-activation voltage of KCa. Data are grouped by as they are in **B**. Statistical comparisons were performed as in **B**.

a different path in the space of activity profiles and intrinsic properties. Ultimately, this resulted in distinct final activity patterns, all of which were consistent with the $Ca^{2+}$ targets.

## Half-(in)activation alterations contribute to activity homeostasis

This model also simulates how the assembled bursters recover their activity following perturbations. In particular, we modeled an increase in extracellular potassium concentration by altering both the potassium reversal potential and the leak reversal potential (see Methods). For these simulations, we used the previously tuned models SP1–SP20. They begin with the homeostatic mechanism engaged but idling because these models already met the prespecified activity targets. However, once extracellular potassium concentration increases, the shifts in reversal potentials cause all 20 models to experience depolarization block. In response, the homeostatic mechanism becomes operative and adjusts the neuron's maximal conductances and half-(in)activations.

We anticipated that the perturbation would depolarize the membrane. To accommodate this shift, we increased the allowable deviation for the DC calcium sensor, $\Delta_D$ (see Methods), from 0.015 to 0.035. *Figure 5A* shows SP1's activity before perturbation (blue) and after 265 min of recovery (red), under three conditions: fast half-(in)activation regulation, intermediate (same timescale as conductance regulation), and no half-(in)activation regulation. The depolarized membrane potential is evident in the more positive maximum hyperpolarizations during perturbation. Notably, although all three cases eventually re-stabilized to activity patterns that satisfied the calcium targets, the resulting burst patterns were different.

SP1 was one of eleven models that successfully recovered under all three homeostatic conditions. Eight models required some form of half-(in)activation modulation to stabilize; in these cases, conductance regulation alone was not sufficient for recovery. Among these eight, two only recovered when half-(in)activation changes were fast, and another two only recovered when they were slow. Evidently, for some models, the ability to recover bursting activity depended on both the presence and speed of half-(in)activation modulation.

*Figure 5B, C* quantifies the variability in post-recovery activity patterns and intrinsic parameters across different half-(in)activation regulation timescales. Comparing the fast ($\tau_{half} = 6$ s, blue) and intermediate ($\tau_{half} = 600$ s, green) conditions reveals significant differences in burst period, interburst interval, and spike height (*Figure 5B*). Notably, these same features also showed significant variation in *Figure 3A*, where timescale-dependent effects were observed during the initial assembly of bursters. Similarly, the half-activation voltage of the KCa current differs significantly between these two timescales (*Figure 5C*), paralleling the differences observed in *Figure 3C*. In models that successfully recovered from perturbation, the timescale of voltage-dependence modulation influenced both the resulting activity patterns and the intrinsic parameters of the stabilized bursters.

## Discussion
### Timescale of channel voltage-dependence alterations

While alterations in channel voltage dependence are known to modulate a neuron's excitability (*Levitan, 1988*), their role in stabilizing a neuron around an activity target has not been extensively explored. This study explores how modifying the *timescale* of voltage-dependence regulation influences both the initial attainment of a target activity pattern and its maintenance during perturbation. While all models met the activity target regardless of whether voltage-dependence modulation was present, recovery after perturbation depended on both the presence and timescale of this mechanism. Furthermore, in both scenarios, the timescale shaped the stabilized neuron's specific activity

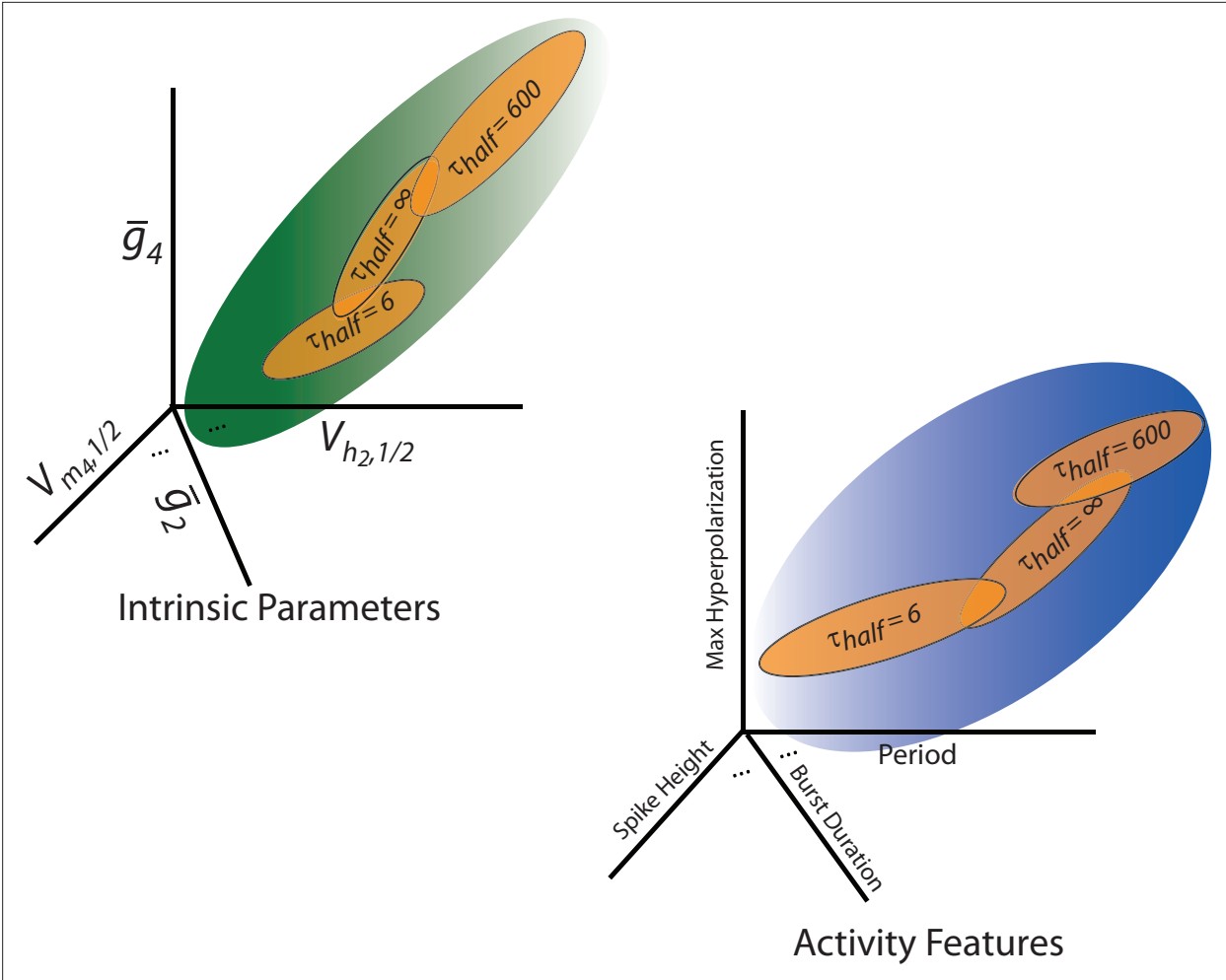

**Figure 6.** The impact of adjusting the timescale of ion channel (in)activation curve alterations can be illustrated conceptually using the space of activity characteristics (right) and underlying intrinsic parameters (left). The regions in green and blue are the activity patterns that are consistent with the $Ca^{2+}$ targets in the space of activity characteristics and underlying intrinsic parameters, respectively. As the rate of ion channel half-(in)activation adjustments in response to deviations from $Ca^{2+}$ targets is changed, the regions targeted by the model also shift (shown in orange). Regardless of the timescale, all targeted regions reside within a larger region encompassing all activity profile measurements and intrinsic parameters consistent with the $Ca^{2+}$ targets.

characteristics and underlying intrinsic parameters. This suggests a general role for the timescale of half-(in)activation modulation in sculpting stable activity states.

*Figure 6* provides a conceptual summary of this idea. The space of possible intrinsic parameter configurations (*Figure 6A*) and electrical activity patterns (*Figure 6B*) each contain regions that satisfy a given activity target—depicted in blue for activity characteristics and green for intrinsic parameters. The perturbations administered in this study simply shift these target-satisfying regions. When channel gating properties are altered quickly in response to deviations from the target activity, the neuron ultimately settles into a stable activity pattern. The resulting electrical patterns are shown in *Figure 6* as the orange bubble labeled '$\tau_{half} = 6$ s'. At a slower timescale of alterations, the activity characteristics and model parameters of the electrical patterns shift, as illustrated by the other orange subregions in *Figure 6*. This shift conceptually highlights how changing the timescale of alterations to channel voltage dependence can move the neuron through different regions within the space of activity characteristics or intrinsic parameters while still maintaining its activity target.

Our simulations show that altering the timescale of the channels' voltage-dependent gating properties impacts its activity characteristics. However, what biological processes can implement these alterations? Fast timescale post-translational alterations can be implemented by second messengers that are anchored next to ion channels (*Fraser and Scott, 1999*; *Johnson et al., 1994*) or second messengers that diffuse across the cytosol (*Agarwal et al., 2016*; *Zaccolo et al., 2006*). Phosphorylation by

kinases is a commonly studied post-translational modification; however, there are other modifications that also occur on these timescales, such as *S*-nitrosylation (*Broillet, 1999*). On slower timescales, the neuron can modulate the transcription and translation of auxiliary subunits based on intracellular calcium levels. For instance, Potassium Channel-Interacting Proteins are known to be controlled by CREB (*Wu et al., 2023*).

## Biological relevance

One application for the simulations involving the self-assembly of activity may be to model the initial phases of neural development, when a neuron transitions from having little or no electrical activity to possessing it (*Baccaglini and Spitzer, 1977*). As shown in *Figure 6*, the timescale of (in)activation curve alterations defines a neuron's activity characteristics and intrinsic properties. As such, neurons may actively adjust these timescales to achieve a specific electrical activity aligned with a developmental phase's activity targets. Indeed, developmental phases are marked by changes in ion channel density and voltage dependence, leading to distinct electrical activity at each stage (*Baccaglini and Spitzer, 1977*; *Gao and Ziskind-Conhaim, 1998*; *Goldberg et al., 2011*; *Hunsberger and Mynlieff, 2020*; *McCormick and Prince, 1987*; *Moody and Bosma, 2005*; *O'Leary et al., 2014*; *Picken Bahrey and Moody, 2003*).

Additionally, our results show that activity-dependent regulation of channel voltage dependence can play a critical role in restoring neuronal activity during perturbations (*Figure 5*). Specifically, the presence and timing of half-(in)activation modulation influenced whether the model neuron could successfully return to its target activity pattern. Many model neurons only achieved recovery when a half-(in)activation mechanism was present. Moreover, the speed of this modulation shaped recovery outcomes in nuanced ways: some model neurons reached their targets only when voltage dependence was adjusted rapidly, while others did so only when these changes occurred slowly. These observations all suggest that impairments in a neuron's ability to modulate the voltage dependence of its channels may lead to disruptions in activity-dependent homeostasis. This may have implications for conditions such as addiction (*Kourrich et al., 2015*) and Alzheimer's disease (*Styr and Slutsky, 2018*), where disruptions in homeostatic processes are thought to contribute to pathogenesis.

In conclusion, our findings suggest that when a neuron begins from a state of no activity and small randomly specified channel densities, changes in channel density are essential for attaining activity. In this context, activity-dependent changes in channel voltage dependence alone did not assemble bursting from these low-conductance initial states (*Figure 1B*). Based on this evidence alone, one might conclude that shifts in (in)activation curves have negligible impact, effectively being overshadowed by the larger changes in excitability driven by alterations in channel density. This perspective aligns with a common assumption in systems analysis: that short- and long-timescale processes are sufficiently distinct and can be studied independently without significant interaction. However, our results demonstrate that fast shifts in half-(in)activation constrain the types of activity characteristics and intrinsic properties a neuron ultimately expresses (*Figure 6*). These fast alterations, while subtle, manifest over long timescales required to assemble activity. This study exemplifies how interactions between processes operating on different timescales can influence one another. Such principles are particularly relevant for understanding multiscale interactions in neuroscience, such as the connections between subcellular, cellular, and network-level properties (*Bhalla, 2014*; *Marom, 2010*).

## Methods
### Details of the model and its implementation

Appendix 1 presents a scheme for the equations used in the model: a Hodgkin–Huxley type equation with seven intrinsic currents, $Ca^{2+}$ sensors, and an activity-dependent regulation mechanism.

The Hodgkin–Huxley type neuron consisted of seven intrinsic currents (*Equations A1–A3*) and an intracellular $Ca^{2+}$ concentration calculation (*Equation A4*). The neuron's capacitance ($C$) was set to 1 nF. The currents were labeled by the index $i$ and were in order: fast sodium ($I_{Na}$), transient calcium ($I_{CaT}$), slow calcium ($I_{CaS}$), hyperpolarization-activated inward cation current ($I_H$), potassium rectifier ($I_{Kd}$), calcium-dependent potassium ($I_{KCa}$), and fast transient potassium ($I_A$). Additionally, there was a leak current ($I_L$). The maximal conductances were given by $\bar{g}_i$. Activation and inactivation dynamics were given by $m_i$ and $h_i$, respectively. The exponents $q_i$ were given by 3,3,3,1,4,4,3, respectively.

The equilibrium/reversal potentials of the sodium current, hyperpolarization-activated cation current, potassium currents, and leak current were +30, –20, –80, and –50 mV, respectively. The calcium equilibrium potential was computed via the Nernst equation (*Equation A5*) using the computed internal Ca²⁺ concentration ([Ca²⁺]ᵢ) and an external Ca²⁺ concentration of 3 mM (temperature was 10°C).

We modeled perturbations in neuronal activity by a 2.5-fold increase in extracellular potassium concentration. Using the Nernst equation, this shift corresponds to a change in the potassium reversal potential from −80 to −55 mV. We assume that leak current in our model includes a potassium component, so that the perturbation also impacts the leak reversal potential. Additionally, assuming that the remaining leak current comes from sodium channels, we can compute how to perturb the leak reversal potential. The leak maximal conductance for our model is fixed at 0.01 μS. The new leak reversal potential was calculated using standard computation involving the conductance-weighted average of potassium and sodium reversal potentials, yielding a value of approximately –32 mV.

The activation curves and inactivation curves were given by $m_{i,\infty}$ and $h_{i,\infty}$. Their associated time constants were $\tau_{m_i}$ and $\tau_{h_i}$, respectively. The (in)activation curves and their associated time constants were functions of the membrane voltage, $V$ (see tables in Appendix 1, Neuron model). Note that $I_{KCa}$ had an activation curve that depends on internal Ca²⁺ concentration. The internal calcium concentration was computed using $I_{CaT}$ and $I_{CaS}$ (*Equation A4*).

There were three Ca²⁺ sensors that responded to different measures associated with fluctuations of [Ca²⁺]ᵢ during a burst. They were used to determine how close these measures are to their targets. These measures were the amount of [Ca²⁺]ᵢ that entered the neuron: as a result of the slow wave of a burst ($S$), as a result of the spiking activity of a burst ($F$), and on average during a burst ($D$). They were computed using *Equations A6–A8* (Appendix 1, Ca²⁺ Sensors). Each sensor had a corresponding target value—$\bar{S}, \bar{F}, \bar{D}$—which represented the target activity pattern.

The model computed moving averages of the mismatch between each sensor and its target, denoted, $E_S$, $E_F$, and $E_D$ (*Equations A9–A11*). These averages were taken over a 2-s window ($\tau_S = 2000$ ms), allowing the model to smooth out short-term fluctuations and capture sustained deviations from the target. Ideally, $E_S$, $E_F$, and $E_D$, are zero; however, ongoing membrane fluctuations, such as bursting or spiking, still created small variations in these measures. Allowable tolerances are specified in the model as, $\Delta_s$, $\Delta_F$, and $\Delta_D$, respectively. The moving averages of the mismatches between the calcium sensors and their targets are compared against the allowable tolerances and combined in *Equation A12* (Appendix 1, Ca²⁺ Sensors) to create a 'match score', $S_F$ (*Equation A12*).

To compute this match score, we adapted a formulation from *Alonso et al., 2023*, who originally used a root-mean-square (RMS) or $L^2$ norm to combine the sensor mismatches. In that approach, each error ($E_S$, $E_F$, and $E_D$) is divided by its allowable tolerance ($\Delta_s$, $\Delta_F$, and $\Delta_D$) to produce a normalized error. These normalized errors are then squared, summed, and square-rooted to produce a single scalar score that reflects how well the model matches the target activity pattern.

In our version, we instead used an $L^8$ norm, which raises each normalized error to the 8th power before summing and taking the 1/8th root. This formulation emphasizes large deviations in any one sensor, making it easier to pinpoint which feature of the activity is limiting convergence. By amplifying outlier mismatches, this approach provided a clearer view of which sensor was driving model mismatch, helping us both interpret failure modes and tune the model's sensitivity by adjusting the tolerances for individual sensor errors.

Although the $L^8$ norm emphasizes large deviations more strongly than the $L^2$ norm, the choice of norm does not fundamentally alter which models can converge—a model that performs well under one norm can also be made to perform well under another by adjusting the allowable tolerances. The biophysical mechanisms by which neurons detect deviations from target activity and convert them into changes in ion channel properties are still not well understood. Given this uncertainty, and the fact that using different norms ultimately should not affect the convergence of a given model, the use of different norms to combine sensor errors is consistent with the broader basic premise of the model: that intrinsic homeostatic regulation is calcium mediated.

*Equations A13 and A14* were used to evaluate whether the match score was sufficient for the model to cease altering maximal conductances and (in)activation curves, indicating that the target had been achieved. In *Equation A14* (Appendix 1, Ca²⁺ sensors), the 'match score' was compared to a preset threshold value the model would accept, $\rho$. This equation returned 1 if the match score is below $\rho$, a 0 if the match score is above $\rho$, and some number in between 0 and 1 if the match score was

close to the threshold, $\rho$. *Equation A13* (Appendix 1, Ca²⁺ sensors) then effectively computed a 2-s time average of *Equation A14*. *Equation A13* was an activity sensor, $\alpha$, that was plotted in *Figure 1*. It took the value zero when all three Ca²⁺ sensors were close to the Ca²⁺ targets but remained close to one otherwise. The value of this activity sensor was used to slow the evolution of maximal conductances and half-(in)activations as the Ca²⁺ sensors reached their targets. It also served as a measure for how well the model satisfied the Ca²⁺ sensors. In this model, $\rho = 0.3$. The 'sharpness' of the threshold at $\rho = 0.3$ was set by $\Delta_\alpha$. We set $\Delta_\alpha = 0.01$ indicating that when the match score, $S_f$, was greater than ~0.32, the model considered the calcium targets satisfied.

Using the information regarding how well the Ca²⁺ sensors were satisfied, the model regulated conductance densities and (in)activation curves. The maximal conductances $\bar{g}_i$, and the ion channel voltage sensitives, as measured by activation curve shifts, $V_{s_i^m}$, and inactivation curve shifts, $V_{s_i^h}$, from their specified locations ($V_{\hat{m}_{i,1/2}^0}$ for activation curves and $V_{\hat{h}_{i,1/2}^0}$ for inactivation curves), were altered by the model in response to deviations from the calcium targets (*Equations A15–A17*). The latter were adjusted on a timescale given by $\tau_{half}$ and the former by $\tau_g$. The index $i$ ran from 1 through 7 for the maximal conductances (*Equation A15*) and half-(in)activations (*Equation A16*). In *Equation A17*, the index only took on a subset of values: $i = 1, 2, 3, 7$—because only some intrinsic currents in this model had inactivation (the associated currents being listed above).

*Equations A16 and A17* were new equations added to the model. They were built similarly to *Equation A15*. In *Equation A15*, the maximal conductances were altered when the calcium sensors didn't match their targets. There are two parts to *Equation A15*: a part that converts calcium sensor mismatches into changes in maximal conductances and a cubic term. The cubic term prevents unbounded growth in the maximal conductances. We set the constant associated with the cubic term, $\gamma$, to $10^{-7}$. The other term adjusted the maximal conductance based on fluctuations of calcium sensors from their target levels. The parameters $A_i$, $B_i$, and $C_i$ reveal how the sensor fluctuations contribute to the changes of each maximal conductance. These values were provided in *Liu et al., 1998*. Importantly, note that this term is multiplied by $\bar{g}_i$, which prevents the maximal conductance from becoming negative and scales the speed of evolution so that smaller values evolve more slowly and larger values evolve more quickly. As such, multiplication by $\bar{g}_i$ effectively induced exponential changes in the maximal conductance, allowing $\bar{g}_i$ to vary over orders of magnitude.

*Equations A16 and A17*, introduced in this paper, enable the adjustment of the ion channel voltage dependence. *Equations A16 and A17* also each have two components: a component that converts calcium sensor mismatches into changes in half-(in)activations and a cubic term. The cubic term prevents unbounded translations. The cubic term's constant, $\hat{\gamma}$, was set to $10^{-5}$. The other term adjusts the half-activations (*Equation A16*) and half-inactivation (*Equation A17*) based on deviation of calcium sensors from their target levels.

Finally, we discuss how the coefficients $A_i$, $B_i$, and $C_i$ in *Equation A15*, and the corresponding parameters in *Equations A16 and A17* were chosen. The same coefficients used to adjust maximal conductances were used for the inactivation curve shifts, $\hat{\hat{A}}_i$, $\hat{\hat{B}}_i$, and $\hat{\hat{C}}_i$. The coefficients of the activation curve shift are their negatives, $\hat{A}_i$, $\hat{B}_i$, and $\hat{C}_i$. To understand this choice, consider a scenario where the neuron is bursting more rapidly than desired. In that case, then $F > \bar{F}$, $S > \bar{S}$, and $D > \bar{D}$. To reduce the excitability of the neuron, we made the depolarizing current, for example, $I_{Na}$, harder to activate, by shifting the activation curve of $I_{Na}$ to a more depolarizing potential. This implies $\frac{dV_{s_1^m}}{dt} > 0$. Therefore, the coefficients $\hat{A}_i$, $\hat{B}_i$, and $\hat{C}_i$ must be non-positive. The coefficients for $I_{Na}$ in *Liu et al., 1998* were given as $A_1 = 1$, $B_1 = 0$, and $C_1 = 0$ and were chosen so that $\frac{d\bar{g}_1}{dt} < 0$, thereby decreasing the maximal conductance of sodium in the same scenario. So, we flipped the signs to give $\hat{A}_1 = -1$, $\hat{B}_1 = 0$, and $\hat{C}_1 = 0$. In this way, only the $F$ Ca²⁺ sensor contributed to the modulation of $I_{Na}$, but the activation curves were now adjusted to promote homeostasis. Furthermore, the model makes it harder to de-inactivate $I_{Na}$ by shifting its inactivation curve to more hyperpolarized values $\left(\frac{dV_{s_1^h}}{dt} < 0\right)$. In the scenario in which the activity sensors are greater than their activity targets, this corresponds to setting $\hat{\hat{A}}_1 = 0, \hat{\hat{B}}_1 = 0,$ and $\hat{\hat{C}}_1 = 0$.

A similar logic can be used to understand how the activation curves of the outward currents were adjusted. In the scenario described above, the neuron was made less excitable by increasing the amount of an outward current, for example $I_A$. This was done by increasing its maximal conductance or moving its activation curve to more hyperpolarized values. *Liu et al., 1998* assigned the coefficients

$A_7 = 0$, $B_7 = -1$, and $C_7 = -1$ to adjust the maximal conductance of $I_A$. In this scenario where all sensors are above their targets, this implies $\frac{d\bar{g}_7}{dt} > 0$. Following the prescription we outlined above, flipping the signs gives $\hat{A}_7 = 0$, $\hat{B}_7 = 1$, and $\hat{C}_7 = 1$. This, in turn, implied $\frac{dV_{s_7}^m}{dt} < 0$—creating hyperpolarizing shifts in $I_A$'s activation curve when the activity is greater than its target, as we expect. In a similar spirit, we assign the coefficients $\hat{A}_7 = 0$, $\hat{B}_7 = -1$, and $\hat{C}_7 = -1$, so that $I_A$'s inactivation curve shifts to more depolarizing potentials $\left( \frac{dV_{s_7}^{sh}}{dt} > 0 \right)$ to make it easier to de-inactivate.

Note, the coefficients in *Liu et al., 1998* do not perfectly correspond with this reasoning. In *Liu et al., 1998*, some coefficients were assigned values to ensure convergence of the model. Still, we followed the prescription outlined above (*Appendix 1—table 7*).

## Starting parameters

The model has 40 ordinary differential equations: 13 equations describe the neuron's electrical activity, 9 equations describe calcium sensor components, and 18 equations alter the maximal conductances and half-(in)activations (Appendix 1). This means we needed to specify 40 starting values for the model. We defined starting parameters (SPs) as the bursters the model assembles from a particular set of randomly chosen starting maximal conductances and half-(in)activations. The maximal conductances were randomly selected between 0.3 and 0.9 nS and half-(in)-activation shifts are randomly offset between –0.5 and +0.5 mV from their specified half-(in)activation values ($V_{\hat{m}_{i,1/2}^0}$ and $V_{\hat{h}_{i,1/2}^0}$ in *Appendix 1—table 2*). Other starting values were $V = -50$ mV and the gating variables of activation and inactivation were randomly selected between 0.2 and 0.3. The starting value of intracellular calcium concentration was set as $\left[ Ca^{2+} \right]_i = 0.4$ µM. *Equations A9–A11* (sensor errors) and *Equation A13* (activity sensor, $\alpha$) were initialized with the value 1. $H_X$ (*Equation A7*) and $M_X$ (*Equation A6*) were initialized to 0 and 1, respectively.

The random initialization of gating variables was not expected to significantly impact the formation of an activity profile that satisfied the activity target. Bursters may exhibit bistability if they are initialized with different gating variables. However, this is for a fixed set of maximal conductances and half-(in)activations. Any bistability in the initially specified profile was likely lost as the model altered the maximal conductances and half-(in)activationss.

The model was integrated using the Tsitouras 5 implementation of Runga–Kutta 4 (5) in Julia's DifferentialEquations.jl package (*Tsitouras, 2011*). The simulation was saved every 0.5 ms regardless of step size (step size of the integrator was adaptive and left unspecified). The maximum number of iterations was set to $10^{10}$ and relative tolerance was set to 0.0001.

## Burster measurements

We defined regular bursters as activity patterns that exhibited groups of spikes separated by regular quiescent gaps in activity. To identify bursters, we measured the following activity characteristics: period, burst duration, interburst interval, spike height, maximal hyperpolarization, and slow wave amplitude. Peaks were identified as local maxima in the activity pattern. We used the 'findpeaks' function in MATLAB on the last 90 s of activity to identify local maxima in the trace, setting the minimum 'prominence' at 2. The latter parameter rejected certain local maxima from counting as peaks. This parameter is one we arrived at by trial and error. *Figure 7B1, B2* shows red arrows where this function identified peaks. Note in *Figure 7B2*, panels A–C have local maxima that are not counted as peaks (there is one local maxima following the sequence of peaks in the burst). The choice of 'prominence' in the 'findpeaks' function was intentionally set to exclude these. To find troughs, we took the negative of the activity profile and found local maxima using the 'findpeaks' function. For this, the same 'prominence' parameter was used over the same time window. The arrows in *Figure 7C* show an example of what was identified by the algorithm as a peak in this case. Troughs were defined as the negative of the identified points. The red arrow in *Figure 7C* shows the maximum hyperpolarization of a cycle obtained using the method described below. The green arrows are the troughs associated with each spike in the burst (*Figure 7C*).

Maximum hyperpolarization was found as follows. First, all troughs were identified in the manner described above. Then, starting at –85 mV, we checked whether there were any troughs below this value. If there were none, we incremented by 1 to –84 mV and checked again if there were any troughs below this value. This process was repeated until troughs were detected. The troughs identified this

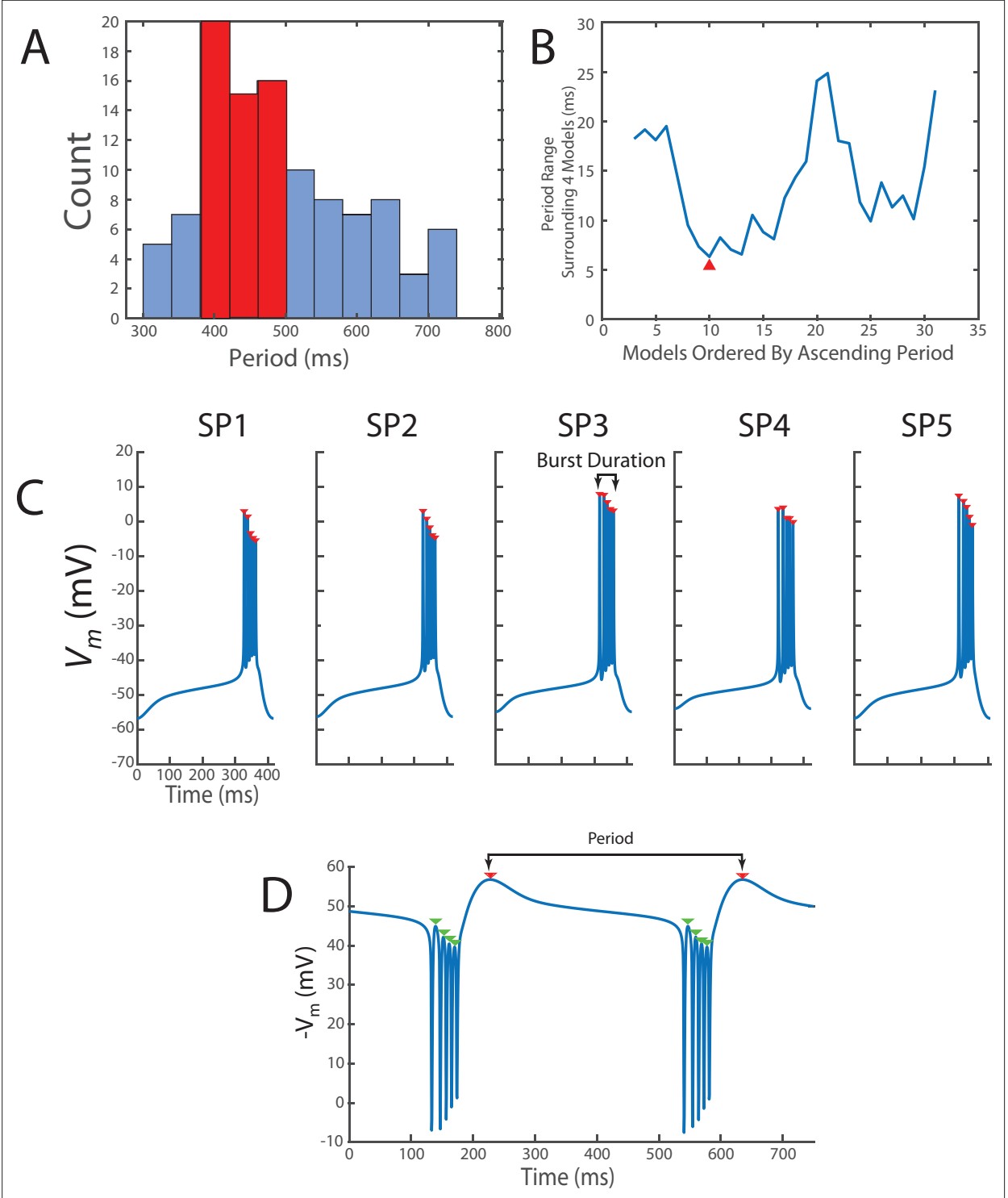

**Figure 7.** We constructed the group of 20 bursters by selecting a putative representative of the entire population and then identifying bursters with similar activity patterns. (**A**) 111 bursters were narrowed to 80 bursters by excluding certain characteristics (see Methods). These models had a period distribution shown in **A**. Highlighted in red are the models chosen for further selection. (**B**) This plot illustrates how closely the periods of the models chosen in **A** align with their two neighbors after they were ordered from smallest to largest. The first model, SP1, was chosen as the neuron with neighbors that also have close periods (red arrow). (**C**) To assemble the group of 20, we symmetrically expanded the inter-instance period range around SP1's period by 20%. SP2–SP5's wave forms are shown. Depolarizing excursions (spikes) were detected using MATLAB's findpeaks function with a prominence threshold of 2 (red arrows). (**D**) Troughs were identified using the findpeaks function on the negative waveform of a cycle period, applying the same prominence threshold. The detected troughs are indicated by red arrows. The example displayed here represents the inverted

*Figure 7 continued on next page*

*Figure 7 continued*

voltage trace of the burster assembled by the model from SP1. The period was measured as the time difference between successive points of maximum hyperpolarization.

way demarcated the ends of a cycle in the activity pattern. They were averaged to obtain the maximum hyperpolarization measurement for the activity pattern.

The time interval between successive maximum hyperpolarizations in the activity was used to measure the cycle period (*Figure 7D*). We averaged these cycle period measurements to give a measurement for the activity pattern's period. To obtain the burst duration, we segmented the activity pattern into cycles using the time of maximum hyperpolarization. The burst duration of a cycle was measured as the time between the first and last spike in the cycle (*Figure 7C*). The interburst interval of the cycle was computed by subtracting the cycle's burst duration from the cycle period.

To measure the slow wave amplitude, we identified the troughs of each cycle. The slow wave amplitude of a cycle is the difference between the maximum hyperpolarization (*Figure 7D*, red arrow) during a cycle and the average of the membrane potential at the troughs associated with each spike in a burst (*Figure 7D*, green arrows). The slow wave amplitudes across all cycles were averaged to obtain a measurement for the activity pattern's slow wave amplitude.

The spike height was obtained by finding the peaks of the membrane potential during each cycle. A measure for the cycle's spike height was obtained by averaging over all peaks, subtracting the cycle's maximum hyperpolarization, then subtracting the cycle's slow wave amplitude. This value was averaged over all cycles to obtain a measure of the activity pattern's spike height.

We distinguished regular bursting from tonic spiking or irregular bursting using the following procedure. We classified an activity pattern as a regular burster when: (1) there was more than 1 spike per cycle, (2) the maximum hyperpolarization measurements for all cycles were within 3 mV of one another, and (3) the spikes per cycle varied by no more than 1 spike. The first condition eliminated tonic spiking, the second condition is a crude measure to check that the average voltage wasn't increasing, and the third checked for irregular bursting. In addition, we verified that the cycle-to-cycle measurements exhibited low variability to ensure the appropriateness of averaging across all cycles. The relative standard deviation for all cycle-to-cycle measurements was less than 3% for the assembled bursters.

## Activity target selection

We needed a set of sensor targets that *consistently* produced regular bursters, regardless of the starting parameters selected. To determine whether a set of calcium targets consistently produced a burster, we monitored the activity sensor $\alpha$. $\alpha$ approached 0 when the calcium sensors were close to their specified targets. There were small oscillations in $\alpha$ that resulted from calcium fluctuations associated with the neuron's activity. These fluctuations were smoothed out by taking a running average over the 8 previous seconds. When a simulation fell below $\alpha = 0.02$ for the last time and remained there for 100 min, we considered the simulation to have '*assembled*' a burster that satisfied the calcium targets. For simulations involving elevated extracellular potassium concentration, this criterion was adjusted: a model was considered to have satisfied the calcium targets if the moving average of $\alpha$ fell below 0.02 and remained there for at least 5 min. This change was made because the perturbation simulations were shorter.

To ensure the activity pattern obtained did not depend on feedback from the activity regulation mechanism, the mechanism was turned off ($\tau_{half} = \infty$, $\tau_g = \infty$) and the simulation ran for an additional 9000 s to ensure $\alpha < 0.02$ for the duration. This step was taken in the initial generation of bursters (111 in total, described below), but not for the analysis where the time constants of the half-activation alterations were changed from $\tau_{half} = 6$ s.

'*Consistently*', in this context, meant that the model assembled bursters from over 95% of the randomly chosen starting parameters. The remaining simulations did not complete assembly within the given simulation period. For the calcium targets we found ($\bar{S} = 0.03$, $\bar{F} = 0.25$, and $\bar{D} = 0.02$), the model 'consistently' produced bursters. To obtain these targets, we tried different combinations of calcium targets until a suitable set of targets consistently produced regular bursters.

## Group of bursters

Using the obtained calcium targets, we assembled a total of 111 regular bursters from a wide range of randomly initialized starting parameters. To create a representative subset of this population, we selected a group of 20 bursters through successive rounds of restriction. The goal was to form a set of models that shared similar burst periods but varied in other activity characteristics and intrinsic properties. This allowed us to assess how different bursters responded to changes in the speed of half-(in)activation alterations as the model attempted to reach or maintain its activity target.

We first filtered the 111 models by slow wave amplitude, retaining only those with amplitudes less than 25 mV. This excluded models with unusually large oscillations and preserved activity patterns with waveforms similar to those shown in *Figure 7C*, reducing the pool to 80 models. The distribution of these bursters' periods is shown in *Figure 7A*. The peak of this distribution is between 380 and 500 ms. We further narrowed the population by selecting bursters around the peak of this distribution (*Figure 7A*, highlighted in red), resulting in 33 bursters (the 'selection group'). This range contained about 30% of the distribution.

In the next round of restriction, we obtained a group of 20. We ordered the bursters from shortest to longest period. For each burster from the 3rd to the 31st position, we examined the two bursters immediately preceding and the two bursters immediately succeeding in this sorted list and computed the range of burst periods across these five bursters (*Figure 7B*). This method provided a rough estimate of the variability in periods among neighboring bursters. The burster exhibiting the smallest inter-instance period variability was selected as the first member of the group of 20 (*Figure 7B*, red arrow). This first burster is referred to as SP1 in *Figure 1* (also shown in *Figure 7C*) and has a period of about 416 ms. In the second step, we symmetrically expanded the inter-instance period range around SP1's period by 20%. We randomly selected an additional 20 models from this group to create the group of 20.

## Model assumptions

This model of channel voltage-dependence alterations is based on certain assumptions. The concept outlined in *Figure 6* is not expected to depend on these choices. However, examining these assumptions reveals additional insights.

The model makes two simplifications on how quickly channel gating properties can be altered. First, the model assumes a single, uniform time constant for modifying all intrinsic currents. In reality, each intrinsic current is likely influenced by a combination of mechanisms. For example, the relative contributions of anchored second messengers and freely diffusing kinases in phosphorylating key residues in L-type calcium channels are not well understood and may well involve both processes (*Weiss et al., 2013*). Consequently, the timescale of modulation for $I_{CaS}$—the current to which L-type calcium channels putatively contribute—likely represents a weighted average of these mechanisms. Similarly, $I_{Kr}$ is modulated by PIP2 (a metabolite of calcium-mediated second messenger pathways) and a beta subunit (*Li et al., 2011*). While the presence or absence of PIP2 can change on a fast timescale, the synthesis of the beta subunit occurs over much slower timescales, resulting in mixed temporal dynamics. These examples illustrate that the balance between fast and slow regulatory mechanisms likely varies among intrinsic currents, complicating the assumption of a single time constant. Second, our framework assumes post-translational modifications are removed as quickly (or slowly) as they are added. This simplification fails to account for processes where rapid calcium increases lead to prolonged downstream effects. For instance, the calcium-mediated second messenger CaMKII undergoes autophosphorylation, enabling a brief calcium signal to induce long-lasting changes in protein activity (*Hudmon and Schulman, 2002*; *Kennedy, 1989*). This autophosphorylation is known to influence intrinsic excitability of neurons (*Sametsky et al., 2009*) and may modulate the voltage-dependence of $Ca_V2$ channels (*Jiang et al., 2008*), which putatively contribute to $I_{CaS}$. Accounting for these processes would require a more complex model capable of capturing asymmetries in modification rates.

Another assumption we make is that there are limits on the extent to which (in)activation curves can shift. This limitation is implemented through the cubic term in *Equations A16 and A17*. These constraints reflect the fact that shifts in activation curves are not unlimited. For example, while cAMP can shift the half-activation of $I_H$, this effect saturates at high cAMP concentrations (*DiFrancesco and Tortora, 1991*). In this study, however, a uniform constraint is applied across all intrinsic currents.

In reality, different intrinsic currents likely have unique limits on how much they can shift, requiring distinct functional forms to model their unique restrictions.

Furthermore, we assumed that activity-dependent alterations in channel voltage dependence are implemented by changing half-(in)activations. We have not, however, considered the (in)activation curve's slope or the gating variable's time constant. In this study, the gating variable's voltage-dependent time constant shifts by the same amount as the half-(in)activation. This preserves how quickly the gating variable responds to changes in voltage—regardless of where the activation curve is centered. However, this coupling between the time constant and the activation curve is not necessarily required. For instance, activity-dependent shifts in voltage-dependent time constants have been observed without concomitant shifts in activation curves (*Thoby-Brisson and Simmers, 2002*). The voltage dependence of the time constant can, in principle, change in other ways as well. Also, we presume that the slopes of the (in)activation curves do not respond to changes in activity. The slope of (in)activation curves defines the steepness of the voltage dependence for ion channel gating, thereby influencing the voltage sensitivity of the transition between states (e.g. from closed to open or from inactivated to de-inactivated). In this model, the slopes of these curves are held constant. Consequently, the voltage ranges remain fixed regardless of shifts in the centering of the activation or inactivation curves. Nonetheless, with certain calcium-activated potassium currents, the center and slope of the activation curve depend on calcium concentration (*Wang and Brenner, 2006*). To the extent that calcium entry during activity initiates downstream signaling processes that alter ion channel function, this observation supports the idea that activity could influence the slope of the $I_{KCa}$ activation curve.

In the model, the same activity targets are used to drive changes in both channel density and voltage dependence. This assumes extensive crosstalk between the calcium pathways posited to be responsible for ion channel insertion, deletion, and post-translational modifications. Nonetheless, the extent to which this crosstalk occurs is not known, although it is known that cells can detect the temporal patterns and specific pathways of calcium entry, leading to specific activity-dependent changes (*Bito et al., 1997*; *Fields et al., 1997*; *Gallin and Greenberg, 1995*). Therefore, it is likely that activity-induced calcium influxes trigger changes along distinct pathways—with some impacting only channel gating properties, some impacting channel density, and some impacting both.

It is not necessarily true that all activity-dependent changes in the properties of intrinsic currents are directly controlled by calcium influx. For example, neurons coregulate $I_A$ and $I_H$ expression, so that a change in one current is associated with a change in the other (*MacLean et al., 2005*). A neuromodulatory environment may also play a role in coordinating ion channel levels (*Khorkova and Golowasch, 2007*). In such cases, one may use this model for the channels that are controlled by activity and augment it by explicitly modeling the remaining mechanisms that are independent of activity.

## Acknowledgements

The authors acknowledge the support of NIMH (R01MH046742), NINDS (R35NS097343), and The Swartz Center for Theoretical Neuroscience at Brandeis University. YM acknowledges valuable discussions with lab members. Also, YM acknowledges assistance with code preparation from Gwendolyn Harris.

## Additional information

### Funding

| Funder | Grant reference number | Author |
| --- | --- | --- |
| National Institute of Mental Health | R01MH046742 | Eve Marder |
| National Institute of Neurological Disorders and Stroke | R35NS097343 | Eve Marder |

| Funder | Grant reference number | Author |
|--------|------------------------|--------|

The funders had no role in study design, data collection, and interpretation, or the decision to submit the work for publication.

## Author contributions

Yugarshi Mondal, Conceptualization, Investigation, Methodology, Writing – original draft, Writing – review and editing; Ronald L Calabrese, Conceptualization, Formal analysis, Supervision, Investigation, Visualization, Methodology, Writing – review and editing; Eve Marder, Conceptualization, Supervision, Funding acquisition, Investigation, Writing – review and editing

## Author ORCIDs

Yugarshi Mondal ⓘ https://orcid.org/0000-0002-7620-8436
Ronald L Calabrese ⓘ https://orcid.org/0000-0001-7135-3469
Eve Marder ⓘ https://orcid.org/0000-0001-9632-5448

Reviewer #1 (Public review): https://doi.org/10.7554/eLife.105842.4.sa1
Reviewer #2 (Public review): https://doi.org/10.7554/eLife.105842.4.sa2
Reviewer #3 (Public review): https://doi.org/10.7554/eLife.105842.4.sa3
Author response https://doi.org/10.7554/eLife.105842.4.sa4

# Additional files

## Supplementary files

MDAR checklist

## Data availability

The code and parameters required to reproduce the results in this work are accessible and at Zenodo. Repository: https://github.com/YugarshiM/fastTimescaleHomeostasis (copy archived at *Mondal, 2024*) and DOI: https://doi.org/10.5281/zenodo.12585617.

The following dataset was generated:

| Author(s) | Year | Dataset title | Dataset URL | Database and Identifier |
|-----------|------|---------------|-------------|-------------------------|
| Yugarshi M | 2024 | YugarshiM/ fastTimescaleHomeostasis: createBursters | https://doi.org/ 10.5281/zenodo. 12585617 | Zenodo, 10.5281/ zenodo.12585617 |

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

# Appendix 1

## Model components

### Neuron model

- Hodgkin–Huxley representation of a single compartment neuron containing voltage- and Ca-dependent currents plus leak.
- Basic [Ca²⁺]ᵢ handling.

### Ca²⁺ sensors

- Three different sensors that reflect [Ca²⁺]ᵢ associated with the slow wave of a burst (S), spiking activity of a burst (F), and the average across bursts (D).
- Targets are set for each sensor independently.
- The three different sensor values are compared to their target values and merged into a single score (α) that reflects overall target satisfaction.

### Homeostatic mechanisms

- Maximal conductances are adjusted on the timescale $\tau_g$, with the rate scaled by α so that adjustments slow as sensor targets are satisfied. Similarly, half-(in)activation voltages are adjusted on the timescale $\tau_{half}$ then scaled by α.

## Neuron model

$$C\frac{dV}{dt} = -\sum_{i=1}^{7} \bar{g}_i m_i^{q_i} h_i \left(V - E_i\right) - g_L \left(V - E_L\right) + I_{app}$$, (A1)

$$\frac{dm_i}{dt} = \frac{m_{i,\infty}\left(V - V_{s_i^m}\right) - m_i}{\tau_{m_i}\left(V - V_{s_i^m}\right)}$$, (A2)

$$\frac{dh_i}{dt} = \frac{h_{i,\infty}\left(V - V_{s_i^h}\right) - h_i}{\tau_{h_i}\left(V - V_{s_i^h}\right)}$$, (A3)

$$\frac{d\left[Ca^{2+}\right]_i}{dt} = \frac{1}{20}\left(-.94\ \left(I_{CaT} + I_{CaS}\right) - \left[Ca^{2+}\right]_i + .05\right)$$, (A4)

$$E_{Ca}\left(\left[Ca^{2+}\right]_i\right) = 12.197 * log\left(\frac{\left[Ca^{2+}\right]_i}{3000.0}\right)$$. (A5)

$I_{CaT}$ and $I_{CaS}$ are the $i = 2$ and $i = 3$ parts of the sum in **Equation A1**.

**Appendix 1—table 1.** Activation curve exponents and equilibrium potentials.

| Current | Index ($i$) | $q_i$ | E (mV) |
|---|---|---|---|
| $I_{Na}$ | 1 | 3 | 30 |
| $I_{CaT}$ | 2 | 3 | [set by Nernst equation—**Equation A5**] |

*Appendix 1—table 1 Continued on next page*

*Appendix 1—table 1 Continued*

| Current | Index ($i$) | $q_i$ | $E$ (mV) |
|---|---|---|---|
| $I_{CaS}$ | 3 | 3 | [set by Nernst equation—***Equation A5***] |
| $I_{H}$ | 4 | 1 | −20 |
| $I_{Kd}$ | 5 | 4 | −80 |
| $I_{KCa}$ | 6 | 4 | −80 |
| $I_{A}$ | 7 | 3 | −80 |

**Appendix 1—table 2.** Normative values of half-activation and half-(in)activation used to generate all starting parameter sets by randomized shifts.

Final parameter sets are converted to the normative value minus the shift.

| Current | Index ($i$) | $V_{\hat{m}^0_{i,1/2}}$ (mV) | $V_{\hat{h}^0_{i,1/2}}$ (mV) |
|---|---|---|---|
| $I_{Na}$ | 1 | −25.5 | −48.9 |
| $I_{CaT}$ | 2 | −27.1 | −32.1 |
| $I_{CaS}$ | 3 | −33 | −60 |
| $I_{H}$ | 4 | −70 | |
| $I_{Kd}$ | 5 | −12.3 | |
| $I_{KCa}$ | 6 | −28.3 | |
| $I_{A}$ | 7 | −27.2 | −56.9 |

**Appendix 1—table 3.** Activation curves and associated time constants.

| Current | $m_{i,\infty}$ (mV) | $\tau_{m_i}$ (ms) |
|---|---|---|
| $I_{Na}$ | $\dfrac{1}{1 + \exp\left(\dfrac{V - \left(V_{\hat{m}^0_{1,1/2}} + V_{s^m_1}\right)}{-5.29}\right)}$ | $1.32 - \dfrac{1.26}{1 + \exp\left(\dfrac{\left(V - V_{s^m_1}\right) + 120}{-25}\right)}$ |
| $I_{CaT}$ | $\dfrac{1}{1 + \exp\left(\dfrac{V - \left(V_{\hat{m}^0_{2,1/2}} + V_{s^m_2}\right)}{-7.2}\right)}$ | $21.7 - \dfrac{21.3}{1 + \exp\left(\dfrac{\left(V - V_{s^m_2}\right) + 68.1}{-20.5}\right)}$ |
| $I_{CaS}$ | $\dfrac{1}{1 + \exp\left(\dfrac{V - \left(V_{\hat{m}^0_{3,1/2}} + V_{s^m_3}\right)}{-8.1}\right)}$ | $1.4 + \dfrac{7}{\exp\left(\dfrac{\left(V - V_{s^m_3}\right) + 27}{10}\right) + \exp\left(\dfrac{\left(V - V_{s^m_3}\right) + 70}{-13}\right)}$ |
| $I_{H}$ | $\dfrac{1}{1 + \exp\left(\dfrac{V - \left(V_{\hat{m}^0_{4,1/2}} + V_{s^m_4}\right)}{6}\right)}$ | $272 - \dfrac{-1499}{1 + \exp\left(\dfrac{\left(V - V_{s^m_4}\right) + 42.2}{-8.73}\right)}$ |
| $I_{Kd}$ | $\dfrac{1}{1 + \exp\left(\dfrac{V - \left(V_{\hat{m}^0_{5,1/2}} + V_{s^m_5}\right)}{-11.8}\right)}$ | $7.2 - \dfrac{6.4}{1 + \exp\left(\dfrac{\left(V - V_{s^m_5}\right) + 28.3}{-19.2}\right)}$ |

*Appendix 1—table 3 Continued on next page*

*Appendix 1—table 3 Continued*

| Current | $m_{i,\infty}$ (mV) | | $\tau_{m_i}$ (ms) |
|---|---|---|---|
| $I_{KCa}$ | $\dfrac{1}{1 + \exp\left(\dfrac{V - \left(V_{\hat{m}^0_{6,1/2}} + V_{s^m_6}\right)}{-12.6}\right)}$ | $\dfrac{[Ca^{2+}]_i}{[Ca^{2+}]_i + 3}$ | $90.3 - \dfrac{75.1}{1 + \exp\left(\dfrac{\left(V - V_{s^m_6}\right) + 46}{-22.7}\right)}$ |
| $I_A$ | $\dfrac{1}{1 + \exp\left(\dfrac{V - \left(V_{\hat{m}^0_{7,1/2}} + V_{s^m_7}\right)}{-8.7}\right)}$ | | $11.6 - \dfrac{10.4}{1 + \exp\left(\dfrac{\left(V - V_{s^m_7}\right) + 32.9}{-15.2}\right)}$ |

**Appendix 1—table 4.** Inactivation curves and associated time constants.

| Current | $h_{i,\infty}$ (mV) | $\tau_{h_i}$ (ms) |
|---|---|---|
| $I_{Na}$ | $\dfrac{1}{1 + \exp\left(\dfrac{V - \left(V_{\hat{h}^0_{1,1/2}} + V_{s^h_1}\right)}{5.18}\right)}$ | $\left(1.5 - \dfrac{-1}{1 + \exp\left(\dfrac{\left(V - V_{s^h_1}\right) + 34.9}{3.6}\right)}\right)\left(\dfrac{0.67}{1 + \exp\left(\dfrac{\left(V - V_{s^h_1}\right) + 62.9}{-10}\right)}\right)$ |
| $I_{CaT}$ | $\dfrac{1}{1 + \exp\left(\dfrac{V - \left(V_{\hat{h}^0_{2,1/2}} + V_{s^h_2}\right)}{5.5}\right)}$ | $105 - \dfrac{89.8}{1 + \exp\left(\dfrac{\left(V - V_{s^h_2}\right) + 55.0}{-16.9}\right)}$ |
| $I_{CaS}$ | $\dfrac{1}{1 + \exp\left(\dfrac{V - \left(V_{\hat{h}^0_{3,1/2}} + V_{s^h_3}\right)}{6.2}\right)}$ | $60 + \dfrac{150}{\exp\left(\dfrac{\left(V - V_{s^h_3}\right) + 55}{9}\right) + \exp\left(\dfrac{\left(V - V_{s^h_3}\right) + 65}{-16}\right)}$ |
| $I_H$ | 1 | 1 |
| $I_{Kd}$ | 1 | 1 |
| $I_{KCa}$ | 1 | 1 |
| $I_A$ | $\dfrac{1}{1 + \exp\left(\dfrac{V - \left(V_{\hat{h}^0_{7,1/2}} + V_{s^h_7}\right)}{4.9}\right)}$ | $38.6 - \dfrac{29.2}{1 + \exp\left(\dfrac{\left(V - V_{s^h_7}\right) + 38.9}{-26.5}\right)}$ |

## Ca²⁺ sensors

$$\frac{dM_X}{dt} = \frac{\bar{M}_X \left(I_{CaT} + I_{CaS}\right) - M_X}{\tau_{M_X}} \; ; X = F, S, D, \tag{A6}$$

$$\frac{dH_X}{dt} = \frac{\bar{H}_X \left(I_{CaT} + I_{CaS}\right) - H_X}{\tau_{H_X}} \; ; X = F, S, D, \tag{A7}$$

$$F = G_F M_F^2 H_F \quad S = G_S M_S^2 H_S \quad D = G_D M_D^2, \tag{A8}$$

$$\frac{dE_F}{dt} = \frac{\left(F - \bar{F}\right) - E_F}{\tau_S}, \tag{A9}$$

$$\frac{dE_S}{dt} = \frac{(S - \bar{S}) - E_S}{\tau_S},$$ (A10)

$$\frac{dE_D}{dt} = \frac{(D - \bar{D}) - E_D}{\tau_S},$$ (A11)

$$S_f = e^{-\left(\left(\frac{E_F}{\Delta_F}\right)^8 + \left(\frac{E_D}{\Delta_D}\right)^8 + \left(\frac{E_S}{\Delta_S}\right)^8\right)^{1/8}},$$ (A12)

$$\tau_\alpha \frac{d\alpha}{dt} = \alpha_\infty (S_f) - \alpha,$$ (A13)

$$\alpha_\infty (S_f) = \frac{1}{1 + e^{\frac{-(S_f + \rho)}{\Delta_\alpha}}}.$$ (A14)

$G_F, G_S$, and $G_D$ are set to 53, 3, and 1, respectively. Also, $\Delta_F = 0.1$, $\Delta_S = 0.008$, and $\Delta_D = 0.015$. During simulations involving increased extracellular potassium concentration $\Delta_D$ is increased to 0.035.

**Appendix 1—table 5.** $M_X$ parameters and time constants.

|   | Z | $\tau_{M_X}$ |
|---|---|---|
| $F$ | 14.8 | 0.5 |
| $S$ | 7.2 | 50 |
| $D$ | 3 | 500 |

$\bar{M}_X$ is a sigmoid function with centering parameter $Z$:

$$\bar{M}_X = \frac{1}{1 + \exp\left(Z + (I_{CaT} + I_{CaS})\right)}.$$

**Appendix 1—table 6.** $H_X$ parameters and time constants.

|   | Z | $\tau_{H_X}$ |
|---|---|---|
| $F$ | 9.8 | 1.5 |
| $S$ | 2.8 | 60 |
| $D$ | – | – |

$\bar{H}_X$ is a sigmoid function with centering parameter $Z$:

$$\bar{H}_X = \frac{1}{1 + \exp\left(-Z - (I_{CaT} + I_{CaS})\right)}.$$

## Homeostatic mechanisms

$$\frac{\tau_g}{\alpha} \frac{d\bar{g}_i}{dt} = \left[A_i\left(\bar{F} - F\right) + B_i\left(\bar{S} - S\right) + C_i\left(\bar{D} - D\right)\right]\bar{g}_i - \gamma\bar{g}_i^3,$$ (A15)

$$\frac{\tau_{half}}{\alpha} \frac{dV_{s_i^m}}{dt} = \left[\hat{A}_i(\bar{F} - F) + \hat{B}_l(\bar{S} - S) + \hat{C}_l(\bar{D} - D)\right] - \hat{\gamma}\left(V_{s_i^m}\right)^3,$$ (A16)

$$\frac{\tau_{half}}{\alpha} \frac{dV_{s_i^h}}{dt} = \left[\hat{\hat{A}}_i\left(\bar{F} - F\right) + \hat{\hat{B}}_i\left(\bar{S} - S\right) + \hat{\hat{C}}_i\left(\bar{D} - D\right)\right] - \hat{\gamma}\left(V_{s_i^h}\right)^3.$$ (A17)

**Appendix 1—table 7.** Parameters describing homeostatic modulation of intrinsic current properties (Li matrix).

| Current | Index ($i$) | $A_i$ | $B_i$ | $C_i$ | $\hat{A}_i$ | $\hat{B}_i$ | $\hat{C}_i$ | $\hat{\hat{A}}_i$ | $\hat{\hat{B}}_i$ | $\hat{\hat{C}}_i$ |
|---|---|---|---|---|---|---|---|---|---|---|
| $I_{Na}$ | 1 | +1 | 0 | 0 | −1 | 0 | 0 | +1 | 0 | 0 |
| $I_{CaT}$ | 2 | 0 | +1 | 0 | 0 | −1 | 0 | 0 | +1 | 0 |
| $I_{CaS}$ | 3 | 0 | +1 | 0 | 0 | −1 | 0 | 0 | +1 | 0 |
| $I_{H}$ | 4 | +1 | −1 | 0 | −1 | +1 | 0 | +1 | −1 | 0 |
| $I_{Kd}$ | 5 | 0 | −1 | −1 | 0 | +1 | +1 | 0 | −1 | −1 |
| $I_{KCa}$ | 6 | 0 | −1 | −1 | 0 | +1 | +1 | 0 | −1 | −1 |
| $I_{A}$ | 7 | 0 | +1 | +1 | 0 | −1 | −1 | 0 | +1 | +1 |

In this paper, the values $\hat{A}_i$, $\hat{B}_i$, and $\hat{C}_i$ are the negative of $A_i$, $B_i$, and $C_i$, respectively. The values $\hat{\hat{A}}_i$, $\hat{\hat{B}}_i$, and $\hat{\hat{C}}_i$ are equal to $A_i$, $B_i$, and $C_i$, respectively. The manner at which we arrive at these values is outlined in Methods. As outlined in the Discussion, these coefficients do not need to take these values.

