## [Editor Report · eLife Assessment]

This **important** computational study investigates homeostatic plasticity mechanisms that neurons may employ to achieve and maintain stable target activity patterns. The work extends previous analyses of calcium-dependent homeostatic mechanisms based on ion channel density by considering activity-dependent shifts in channel activation and inactivation properties that operate on faster and potentially variable timescales. The model simulations **convincingly** demonstrate the potential functional importance of these mechanisms.

---

## [Referee Report · Reviewer #1 (Public review)]

This revision of the computational study by Mondal et al addresses several issues that I raised in the previous round of reviews and, as such, is greatly improved. The manuscript is more readable, its findings are more clearly described, and both the introduction and the discussion sections are tighter and more to the point. And thank you for addressing the three timescales of half activation/inactivation parameters. It makes the mechanism clearer.

Some issues remain that I bring up below.

Comment:

I still have a bone to pick with the claim that "activity-dependent changes in channel voltage-dependence alone are insufficient to attain bursting". As I mentioned in my previous comment, this is also the case for the gmax values (channel density). If you choose the gmax's to be in a reasonable range, then the statement above is simply cannot be true. And if, in contrast, you choose the activation/inactivation parameters to be unreasonable, then no set of gmax's can produce proper activity. So I remain baffled what exactly is the point that the authors are trying to make.

---

## [Referee Report · Reviewer #2 (Public review)]

Summary:

In this study, Mondal and co-authors present the development of a computational model of homeostatic plasticity incorporating activity-dependent regulation of gating properties (activation, inactivation) of ion channels. The authors show that, similar to what has been observed for activity-dependent regulation of ion channel conductances, implementing activity-dependent regulation of voltage sensitivity participates in the achievement of a target phenotype (bursting or spiking). The results however suggest that activity-dependent regulation of voltage sensitivity is not sufficient to allow this and needs to be associated with the regulation of ion channel conductances in order to reliably reach target phenotype. Although the implementation of this biologically relevant phenomenon is undeniably relevant, a few important questions are left unanswered.

Strengths:

(1) Implementing activity-dependent regulation of gating properties of ion channels is biologically relevant.

(2) The modeling work appears to be well performed and provides results that are consistent with previous work performed by the same group.

Weaknesses:

(1) The main question not addressed in the paper is the relative efficiency and/or participation of voltage-dependence regulation compared to channel conductance in achieving the expected pattern of activity. Is voltage-dependence participating to 50% or 10%. Although this is a difficult question to answer (and it might even be difficult to provide a number), it is important to determine whether channel conductance regulation remains the main parameter allowing the achievement of a precise pattern of activity (or its recovery after perturbation).

(2) Another related question is whether the speed of recovery is significantly modified by implementing voltage-dependence regulation (it seems to be the case looking at Figure 3). More generally, I believe it would be important to give insights into the overall benefit of implementing voltage-dependence regulation, beyond its rather obvious biological relevance.

(3) Along the same line, the conclusion about how voltage-dependence regulation and channel conductance regulation interact to provide the neuron with the expected activity pattern (summarized and illustrated in Figure 6) is rather qualitative. Consistent with my previous comments, one would expect some quantitative answers to this question, rather than an illustration that approximately places a solution in parameter space.

---

## [Referee Report · Reviewer #3 (Public review)]

Mondal et al. use computational modeling to investigate how activity-dependent shifts in voltage-dependent (in)activation curves can complement changes in ion channel conductance to support homeostatic plasticity. While it is well established that the voltage-dependent properties of ion channels influence neuronal excitability, their potential role in homeostatic regulation, alongside conductance changes, has remained largely unexplored. The results presented here demonstrate that activity-dependent regulation of voltage dependence can interact with conductance plasticity to enable neurons to attain and maintain target activity patterns, in this case, intrinsic bursting. Notably, the timescale of these voltage-dependent shifts influences the final steady-state configuration of the model, shaping both channel parameters and activity features such as burst period and duration. A major conclusion of the study is that altering this timescale can seamlessly modulate a neuron's intrinsic properties, which the authors suggest may be a mechanism for adaptation to perturbations.

While this conclusion is largely well-supported, additional analyses could help clarify its scope. For instance, the effects of timescale alterations are clearly demonstrated when the model transitions from an initial state that does not meet the target activity pattern to a new stable state. However, Fig. 6 and the accompanying discussion appear to suggest that changing the timescale alone is sufficient to shift neuronal activity more generally. It would be helpful to clarify that this effect primarily applies during periods of adaptation, such as neurodevelopment or in response to perturbations, and not necessarily once the system has reached a stable, steady state. As currently presented, the simulations do not test whether modifying the timescale can influence activity after the model has stabilized. In such conditions, changes in timescale are unlikely to affect network dynamics unless they somehow alter the stability of the solution, which is not shown here. That said, it seems plausible that real neurons experience ongoing small perturbations which, in conjunction with changes in timescale, could allow gradual shifts toward new solutions. This possibility is not discussed but could be a fruitful direction for future work.

Editor's note: The authors have adequately addressed the concerns raised in the public reviews above, as well as the previous recommendations, and revised the manuscript where necessary.

---

## [Author Response]

The following is the authors’ response to the previous reviews

**Reviewer #1 (Public review):**
I still have a bone to pick with the claim that "activity-dependent changes in channel voltage-dependence alone are insufficient to attain bursting". As I mentioned in my previous comment, this is also the case for the gmax values (channel density). If you choose the gmax's to be in a reasonable range, then the statement above is simply cannot be true. And if, in contrast, you choose the activation/inactivation parameters to be unreasonable, then no set of gmax's can produce proper activity. So I remain baffled what exactly is the point that the authors are trying to make.

We thank the reviewer for this clarification. We did not intend to imply that voltage-dependence modulation is universally incapable of supporting bursting or that conductance changes alone are universally sufficient. To avoid any overstatement, we now write:

“…activity-dependent changes in channel voltage-dependence alone did not assemble bursting from these low-conductance initial states (cf. Figure 1B)”.

**Reviewer #2 (Public review):**
(1) The main question not addressed in the paper is the relative efficiency and/or participation of voltage-dependence regulation compared to channel conductance in achieving the expected pattern of activity. Is voltage-dependence participating to 50% or 10%. Although this is a difficult question to answer (and it might even be difficult to provide a number), it is important to determine whether channel conductance regulation remains the main parameter allowing the achievement of a precise pattern of activity (or its recovery after perturbation).

We appreciate the reviewer’s interest in a quantitative partitioning of the contributions from voltage-dependence regulation versus conductance regulation. We agree that this would be an important analysis in principle. In practice, obtaining this would be difficult.

Our goal here was to establish the principle: that half-(in)activation shifts can meaningfully influence recovery. This is not an obvious result, given that these two processes can act on vastly different timescales.

That said, our current dataset does provide partial quantitative insight. Eight of the twenty models required some form of voltage-dependence modulation to recover; among these, two only recovered under fast modulation and two only under slow modulation. This demonstrates that voltage-dependence regulation is essential for recovery in some neurons, and its timescale critically shapes the outcome.

(2) Another related question is whether the speed of recovery is significantly modified by implemeting voltage-dependence regulation (it seems to be the case looking at Figure 3). More generally, I believe it would be important to give insights into the overall benefit of implementing voltage-dependence regulation, beyond its rather obvious biological relevance.

Our current results suggest that voltage-dependence regulation can indeed accelerate recovery, as illustrated in Figure 3 and supported by additional simulations (not shown). However, a fully quantitative comparison (e.g., time-to-recovery distributions or survival analysis) would require a much larger ensemble of degenerate models to achieve sufficient statistical power across all four conditions. Generating and simulating this expanded model set is computationally intensive, requiring stochastic searches in a high-dimensional parameter space, full time-course simulations, and a subsequent selection process that may succeed or fail.

The principal aim of the present study is conceptual: to demonstrate that this multi-timescale homeostatic model—built here for the first time—can capture interactions between conductance regulation and voltage-dependence modulation during assembly (“neurodevelopment”) and perturbation. Establishing the conceptual framework and exploring its qualitative behavior were the necessary first steps before pursuing a large-scale quantitative study.

(3) Along the same line, the conclusion about how voltage-dependence regulation and channel conductance regulation interact to provide the neuron with the expected activity pattern (summarized and illustrated in Figure 6) is rather qualitative. Consistent with my previous comments, one would expect some quantitative answers to this question, rather than an illustration that approximately places a solution in parameter space.

We appreciate the reviewer’s interest in a more quantitative characterization of the interaction between voltage-dependence and conductance regulation (Fig. 6). As noted in our responses to Comments 1 and 2, some of the facets of this interaction—such as the ability to recover from perturbations and the speed of assembly—can be measured.

However, fully quantifying the landscape sketched in Figure 6 would require systematically mapping the regions of high-dimensional parameter space where stable solutions exist. In our model, this space spans 18 dimensions (maximal conductances and half‑(in)activations). Even a coarse grid with three samples per dimension would entail over 100 million simulations, which is computationally prohibitive and would still collapse to a schematic representation for visualization.

For this reason, we chose to present Figure 6 as a conceptual summary, illustrating the qualitative organization of solutions and the role of multi-timescale regulation, rather than attempting an exhaustive mapping. We view this figure as a necessary first step toward guiding future, more quantitative analyses.

**Reviewer #3 (Public review):**
Mondal et al. use computational modeling to investigate how activity-dependent shifts in voltage-dependent (in)activation curves can complement changes in ion channel conductance to support homeostatic plasticity. While it is well established that the voltage-dependent properties of ion channels influence neuronal excitability, their potential role in homeostatic regulation, alongside conductance changes, has remained largely unexplored. The results presented here demonstrate that activity-dependent regulation of voltage dependence can interact with conductance plasticity to enable neurons to attain and maintain target activity patterns, in this case, intrinsic bursting. Notably, the timescale of these voltage-dependent shifts influences the final steady-state configuration of the model, shaping both channel parameters and activity features such as burst period and duration. A major conclusion of the study is that altering this timescale can seamlessly modulate a neuron's intrinsic properties, which the authors suggest may be a mechanism for adaptation to perturbations.While this conclusion is largely well-supported, additional analyses could help clarify its scope. For instance, the effects of timescale alterations are clearly demonstrated when the model transitions from an initial state that does not meet the target activity pattern to a new stable state. However, Fig. 6 and the accompanying discussion appear to suggest that changing the timescale alone is sufficient to shift neuronal activity more generally. It would be helpful to clarify that this effect primarily applies during periods of adaptation, such as neurodevelopment or in response to perturbations, and not necessarily once the system has reached a stable, steady state. As currently presented, the simulations do not test whether modifying the timescale can influence activity after the model has stabilized. In such conditions, changes in timescale are unlikely to affect network dynamics unless they somehow alter the stability of the solution, which is not shown here. That said, it seems plausible that real neurons experience ongoing small perturbations which, in conjunction with changes in timescale, could allow gradual shifts toward new solutions. This possibility is not discussed but could be a fruitful direction for future work.

We thank the reviewer for this thoughtful comment and for highlighting an important point about the scope of our conclusions regarding timescale effects. The reviewer is correct that our simulations demonstrate the influence of voltage-dependence timescale primarily during periods of adaptation—when the neuron is moving from an initial, target-mismatched state toward a final target-satisfying state. Once the system has reached a stable solution, simply changing the timescale of voltage-dependent modulation does not by itself shift the neuron’s activity, unless a new perturbation occurs that re-engages the homeostatic mechanism. We have clarified this point in the revised Discussion.

The confusion likely arose from imprecise phrasing in the original text describing Figure 6. Previously, we wrote:

“When channel gating properties are altered quickly in response to deviations from the target activity, the resulting electrical patterns are shown in Figure 6 as the orange bubble labeled 𝝉_𝒉𝒂𝒍𝒇_ = 6 s”.

We have revised this sentence to emphasize that the orange bubble represents the eventual stable state, rather than implying that timescale changes alone drive activity shifts:

”When channel gating properties are altered quickly in response to deviations from the target activity, the neuron ultimately settles into a stable activity pattern. The resulting electrical patterns are shown in Figure 6 as the orange bubble labeled 𝝉_𝒉𝒂𝒍𝒇_ = 6 s”.

**Reviewer #1 (Recommendations for the authors):**
Unless I am missing something, Figure 2 should be a supplement to Figure 1. I would prefer to see panel B in Figure 1 to indicate that the findings of that figure are general. Panel A really is not showing anything useful to the reader.

We appreciate the suggestion to combine Figure 2 with Figure 1, but we believe keeping Figure 2 separate better preserves the manuscript’s flow. Figure 1 illustrates the mechanism in a single model, while Figure 2 presents the population-level summary that generalizes the phenomenon across all models.

Also, I find Figure 6 unnecessary and its description in the Discussion more detracting than useful. Even with the descriptions, I find nothing in the figure itself that clarifies the concept.

We appreciate the reviewer’s feedback on Figure 6. The purpose of this figure is to conceptually illustrate that multiple degenerate solutions can satisfy the calcium target and that the timescale of voltage‑dependence modulation can influence which region of this solution space is accessed during the acquisition of the activity target. Reviewer 3 noted some confusion about this point. We made a small clarifying edit.

At the risk of being really picky, I also don't see the purpose of Figure 7. And I find it strange to plot -Vm just because that's the argument of findpeaks.

We appreciate the reviewer’s comment on Figure 7. The purpose of this figure is to illustrate exactly what the findpeaks function is detecting, as indicated by the red arrows on the traces. For readers unfamiliar with findpeaks, it may not be obvious how the algorithm interprets the waveform. Showing the peaks directly ensures that the measurements used in our analysis align with what one would intuitively expect.

**Reviewer #2 (Recommendations for the authors):**
The writing of the article has been much improved since the last version. It is much clearer, and the discussion has been improved and better addresses the biological foundations and relevance of the study. However, conclusions are rather qualitative, while one would expect some quantitative answers to be provided by the modeling approach.

We appreciate the reviewer’s concern regarding quantification and share this perspective. As noted above, our study is primarily conceptual. Many aspects of the model, such as calcium handling and channel regulation, are parameterized based on incomplete biological data. These uncertainties make robust quantitative predictions difficult, so we focus on qualitative outcomes that are likely to hold independently of specific parameter choices.